# Regulation of local GTP availability controls RAC1 activity and cell invasion

Anna Bianchi-Smiraglia [1,11✉], David W. Wolff [2,11], Daniel J. Marston[3], Zhiyong Deng[2], Zhannan Han[2], Sudha Moparthy[2], Rebecca M. Wombacher[1], Ashley L. Mussell[1], Shichen Shen[4], Jialin Chen[2], Dong-Hyun Yun[2], Anderson O'Brien Cox[2], Cristina M. Furdui [2], Edward Hurley[5], Maria Laura Feltri [5], Jun Qu[4], Thomas Hollis [6], Jules Berlin Nde Kengne[7], Bernard Fongang[8,9], Rui J. Sousa [9], Mikhail E. Kandel [10], Eugene S. Kandel[1], Klaus M. Hahn[3] & Mikhail A. Nikiforov [2✉]

Physiological changes in GTP levels in live cells have never been considered a regulatory step of RAC1 activation because intracellular GTP concentration (determined by chromatography or mass spectrometry) was shown to be substantially higher than the in vitro RAC1 GTP dissociation constant (RAC1-GTP Kd). Here, by combining genetically encoded GTP biosensors and a RAC1 activity biosensor, we demonstrated that GTP levels fluctuating around RAC1-GTP Kd correlated with changes in RAC1 activity in live cells. Furthermore, RAC1 co-localized in protrusions of invading cells with several guanylate metabolism enzymes, including rate-limiting inosine monophosphate dehydrogenase 2 (IMPDH2), which was partially due to direct RAC1-IMPDH2 interaction. Substitution of endogenous IMPDH2 with IMPDH2 mutants incapable of binding RAC1 did not affect total intracellular GTP levels but suppressed RAC1 activity. Targeting IMPDH2 away from the plasma membrane did not alter total intracellular GTP pools but decreased GTP levels in cell protrusions, RAC1 activity, and cell invasion. These data provide a mechanism of regulation of RAC1 activity by local GTP pools in live cells.

[1] Department of Cell Stress Biology, Roswell Park Comprehensive Cancer Center, Buffalo, NY, USA. [2] Department of Cancer Biology, Wake Forest University Baptist Medical Center, Winston-Salem, NC, USA. [3] Department of Pharmacology and Lineberger Cancer Center, University of North Carolina at Chapel Hill, Chapel Hill, NC, USA. [4] New York State Center of Excellence Bioinformatics and Life Sciences, State University of New York at Buffalo, Buffalo, NY, USA. [5] Department of Biochemistry and Neurology, Hunter James Kelly Research Institute, Jacobs School of Medicine and Biomedical Sciences, State University of New York at Buffalo, Buffalo, NY, USA. [6] Department of Biochemistry, Center for Structural Biology, Wake Forest School of Medicine, Winston-Salem, NC, USA. [7] Department of Physics, University of Houston, Houston, TX, USA. [8] Glenn Biggs Institute for Alzheimer's and Neurodegenerative Diseases, University of Texas Health Science Center at San Antonio, San Antonio, TX, USA. [9] Department of Biochemistry and Structural Biology, University of Texas Health Science Center at San Antonio, San Antonio, TX, USA. [10] Groq, 400 Castro St #600, Mountain View, CA 94041, USA. [11] These authors contributed equally Anna Bianchi-Smiraglia, David W. Wolff. ✉email: Anna.Bianchi-Smiraglia@RoswellPark.org; mikhail.nikiforov@duke.edu

Guanine nucleotide-binding proteins (or G-proteins) regulate multiple cellular processes and are observed to be hyper-activated in many human cancers[1]. While a few activating point mutations of Ras-related C3 botulinum toxin substrate 1 (RAC1), a member of the RHO-GTPases family of G-proteins, have been identified in malignancies such as melanoma, breast, and head and neck cancers[2–5], the mechanisms behind the overall over-activation of GTPases in cancer has not yet been elucidated.

In the active, GTP (guanosine triphosphate)-bound, form RAC1 interacts with downstream effectors triggering activation of multiple signaling pathways ultimately resulting in regulation of cell motility, invasion, and progression through the cell cycle[2–6]. On the contrary, GDP-bound RAC1 is inactive. The switch between GTP- and GDP-bound RAC1 is tightly regulated by GTPase-activating proteins (GAPs) that promote GTP hydrolysis and render RAC1 inactive[7,8], and guanine nucleotide exchange factors (GEFs) that promote the release of GDP from RAC1[7,8]. Notably, after releasing GDP from RAC1, GEFs do not increase GTP binding to RAC1, which occurs due to a high intracellular GTP to GDP ratio (~10:1)[9].

Cancer cells often display altered expression of enzymes for the de novo biosynthesis of nucleotides, especially purines, highlighting their dependency on these pathways and substrates[10–16]. In particular, the de novo biosynthesis of guanosine monophosphate (GMP) is driven by the consecutive enzymatic action of inosine monophosphate dehydrogenase 1 and 2 (IMPDH1, 2) and guanosine monophosphate synthase (GMPS), which convert inosine monophosphate (IMP) into xanthosine monophosphate (XMP)[17] and XMP into GMP[18], respectively. GMP reductase (GMPR) is a functional antagonist of IMPDH1/2 and GMPS that reduces GMP to IMP (Supplementary Fig. 1A)[19]. GMP can be further phosphorylated to GTP by members of the nucleoside diphosphate kinase family (NME). Although IMPDH1 and IMPDH2 share 84% amino acid sequence homology[20] and possess comparable enzymatic activity[21], IMPDH1 was demonstrated to constitutively express in normal cells, while IMPDH2 levels were elevated in transformed cells[22].

The role of GTP metabolism in the regulation of G-proteins in live cells remains controversial. Data reported by us and others demonstrated that alterations in guanylate metabolism enzymes levels and/or activity led to changes in the activity of several G-proteins, including members of small RHO-GTPase family[23–27]. On the other hand, physiological changes in GTP levels in live cells had not been thought to affect the activity of G-proteins directly because the concentration of intracellular GTP determined by HPLC (0.5–1.5 mM)[28] is significantly higher than guanylate dissociation constants of the RAS-GTPase family members, which, even in the presence of GEFs does not exceed micromolar range (3–20 μM)[29,30]. However, total intracellular GTP levels determined via chromatography or mass spectroscopy do not reflect the free GTP levels inside the cell, nor reveal subcellular variation in local GTP levels, all of which may have profound effects on local G-protein activation.

Recently, we reported genetically encoded ratiometric fluorescent sensors of free intracellular GTP[31] (Supplementary Fig. 1B). One of the sensors, (termed GEVAL for GTP evaluator) possessed the Keff (GTP concentrations required to obtain 50% of the maximal ratiometric signal) at 32.3 μM GTP as determined in vitro. This sensor, GEVAL30, detects changes in GTP levels of as low as 4 μM and starts getting saturated at ~100 μM, with optimal activity in the 30 μM range[31]. GEVAL30 made it possible to detect GTP fluctuations in live cells at concentrations close to 30 μM[31]. GEVALs were constructed by inserting a cpYFP into a flexible loop region of the bacterial FeoB G-protein that undergoes a GTP-driven conformational change. The binding of GTP to these sensors resulted in a ratiometric change in their fluorescence, thereby providing an internally normalized response to changes in GTP levels[31].

Here, using GEVAL sensors in combination with sensors for RAC1 activity and conventional biochemical, molecular, and cell biology methods, we provide experimental evidence demonstrating that interactions with IMPDH2 and generation of local availability of GTP represent an important mechanism of RAC1 activation in live cells.

## Results

**GTP levels are increased in cell protrusions.** We and others have previously demonstrated that pharmacological or genetic modulation of total intracellular GTP levels alters tumor cell invasion[23–26]. As the formation of cell protrusions (CP) is a key step in the invasion process, we were interested in assessing changes in GTP pools inside the CPs of invading cells. To this end, we combined a previously reported methodology for the induction of CPs in a 3D format[32] with the GEVAL ratiometric fluorescent sensors of free intracellular GTP[31] (Supplementary Fig. 1b). Thus, human breast cancer MDA-MB-231 cells and human melanoma SK-Mel-103 cells were transduced with GEVAL30 or GEVALNull (that cannot bind GTP and serves as a negative control) and plated in serum-free immunofluorescence-compatible media (IFM, see Methods) on top of a transwell with 3 μm pore filter coated with a layer of collagen (Fig. 1a). About 10% FBS-containing IFM was used on the bottom of the well as a chemoattractant. The 3 μm pores allow CP to pass through but prevent the whole cell to do so (Fig. 1a). CPs were induced for 2 h, at which point cells were imaged for GEVAL activity in cell bodies (CBs) and CPs, above and below the filters, respectively (Fig. 1b). Analysis of the Ex405/Ex488 excitation of both GEVAL sensors was performed as previously described[31] to monitor the sensor's activity (and therefore GTP levels). The GEVAL30 activity showed a statistically significant increase in CPs versus CBs (Fig. 1b, c), while there was no significant difference in the activity of GEVALNull between the two compartments (Fig. 1b, c). This indicated that GTP levels (fluctuating around GEVAL30 Keff (~30 μM)) were higher in CPs than in CBs.

To gain mechanistic insight into this phenomenon, we separated CB/CP fractions in MDA-MB-231 and SK-Mel-103 cells via fixation followed by scraping cell material separately off either side of membranes. One of the membranes was stained for actin (phalloidin) and nuclei (Hoechst) after wiping off half of the CB compartment to verify the induction of CPs (Supplementary Fig. 1c). The obtained cell material was analyzed at a ratio of ~1:10/CB:CP in immunoblotting. As expected, RAC1 and integrin β1, which are classical markers of CPs[33,34], were enriched in the CP fraction while the nuclear transcription factor Kruppel like factor 9, KLF9, was enriched in CB fraction (Fig. 1d). Importantly, IMPDH2, GMPS, GMPR, and NME1 were also enriched in CP fraction thus accounting for the observed enrichment of GTP in CPs (Fig. 1b, c).

**GTP levels in CP regulate RAC1 activity and cell invasion.** To evaluate the importance of changes in intracellular GTP for regulation of RAC1 activity and cell invasion, we generated MDA-MB-231 cells with different intracellular localization of IMPDH2. *IMPDH2* mRNA levels in MDA-MB-231 and SK-Mel-103 cells were substantially higher than *IMPDH1* mRNA levels, (Supplementary Fig. 2). In non-transformed cells of the corresponding origin, the *IMPDH2/IMPDH1* levels ratio was less than in their transformed counterparts (Supplementary Fig. 2). Therefore, we focused on IMPDH2 for further studies. First, we depleted MDA-MB-231 cells of endogenous IMPDH2 with a previously validated shRNA[31]. Next, IMPDH2-depleted cells were transduced with

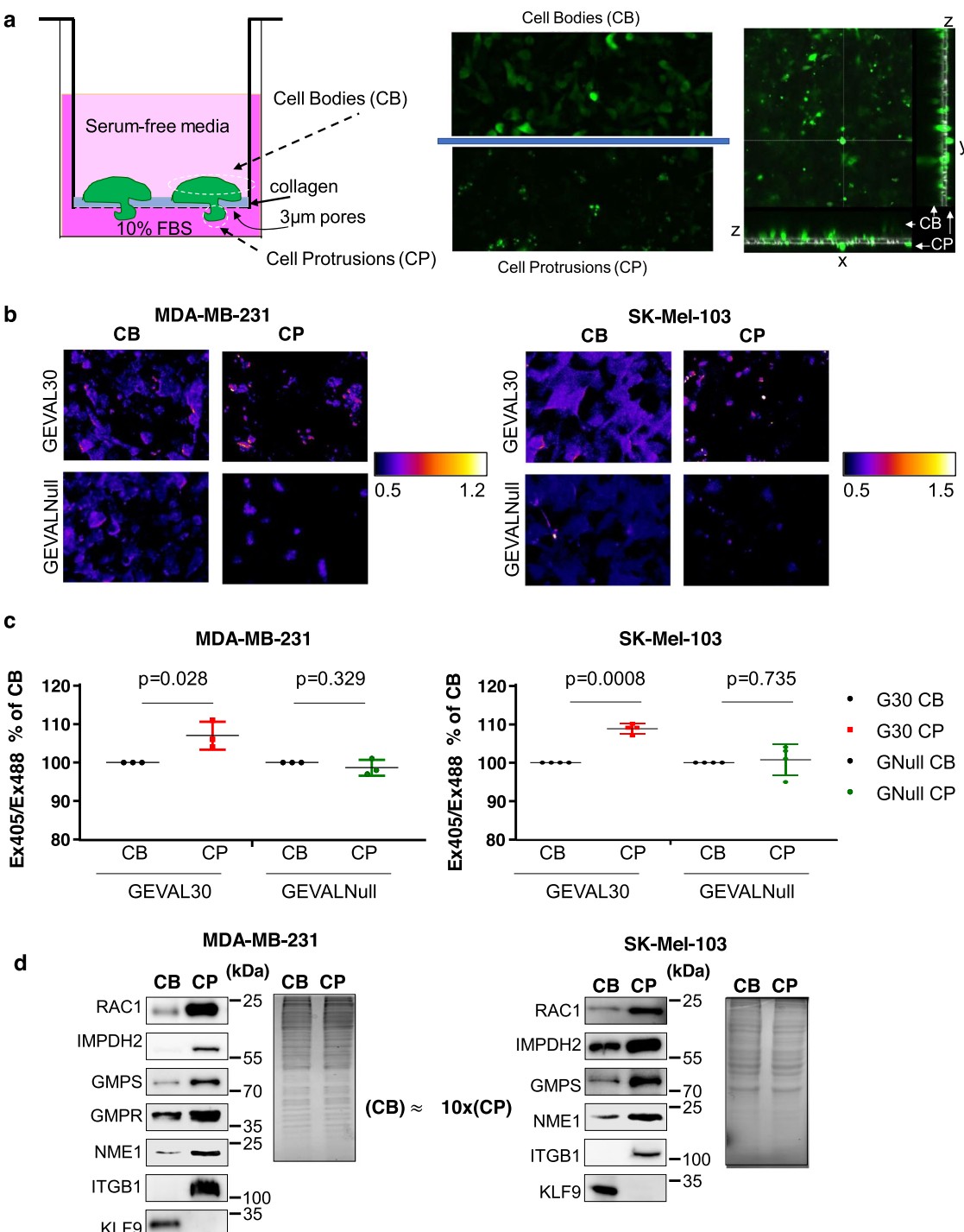

**Fig. 1 Protrusions of invading cells contain increased levels of GTP and guanylate biosynthesis enzymes. a** Schematic representation of the chamber used for separation of cell bodies (CBs) and cell protrusions (CPs) (left), and representative images and orthoslices of CB and CP acquired in one of the two GFP channels (right). **b** Representative false-colored radiometric images of cell bodies and cell protrusions in MDA-MB-231 and SK-Mel-103 cells transduced with GEVAL30 or GEVALNull (GTP Evaluator). **c** Quantification of GEVAL activity in MDA-MB-231 and SK-Mel-103 cells transduced with GEVAL30 (G30) or GEVALNul (GNull). Data is average ± SEM of 3 independent experiments (30 CB and 30 CP analyzed per experiment). Statistics was performed by a two-tailed paired Student's *t*-test. **d** Cell bodies (CB) and cell protrusions (CP) were collected as described in Methods followed by protein quantification and probing of equal amounts of material in immunoblotting with the designated antibodies (left) or staining with Coomassie blue (right). Please note that GMPR is not expressed in SK-Mel-103 cells[27] and was excluded from the analysis. An approximate ratio of CB and CP materials is shown. Shown are representative gel images of at least two independent experiments.

*IMPDH2* ORFs that were modified to include a tag to redirect the resulting IMPDH2 protein to the outer membrane of Golgi (a fragment of the Giantin protein[35]) (IMPDH2-GIA) or to the plasma membrane, using the membrane-targeting sequence of the protein tyrosine kinase LCK[36] (IMPDH2-LCK) (Fig. 2a). The *IMPDH2* ORFs were mutated to become resistant to the shRNA[31]. A cell fractionation assay detected cytoplasmic IMPDH2 localization in all cell populations (Fig. 2a), whereas plasma membrane-

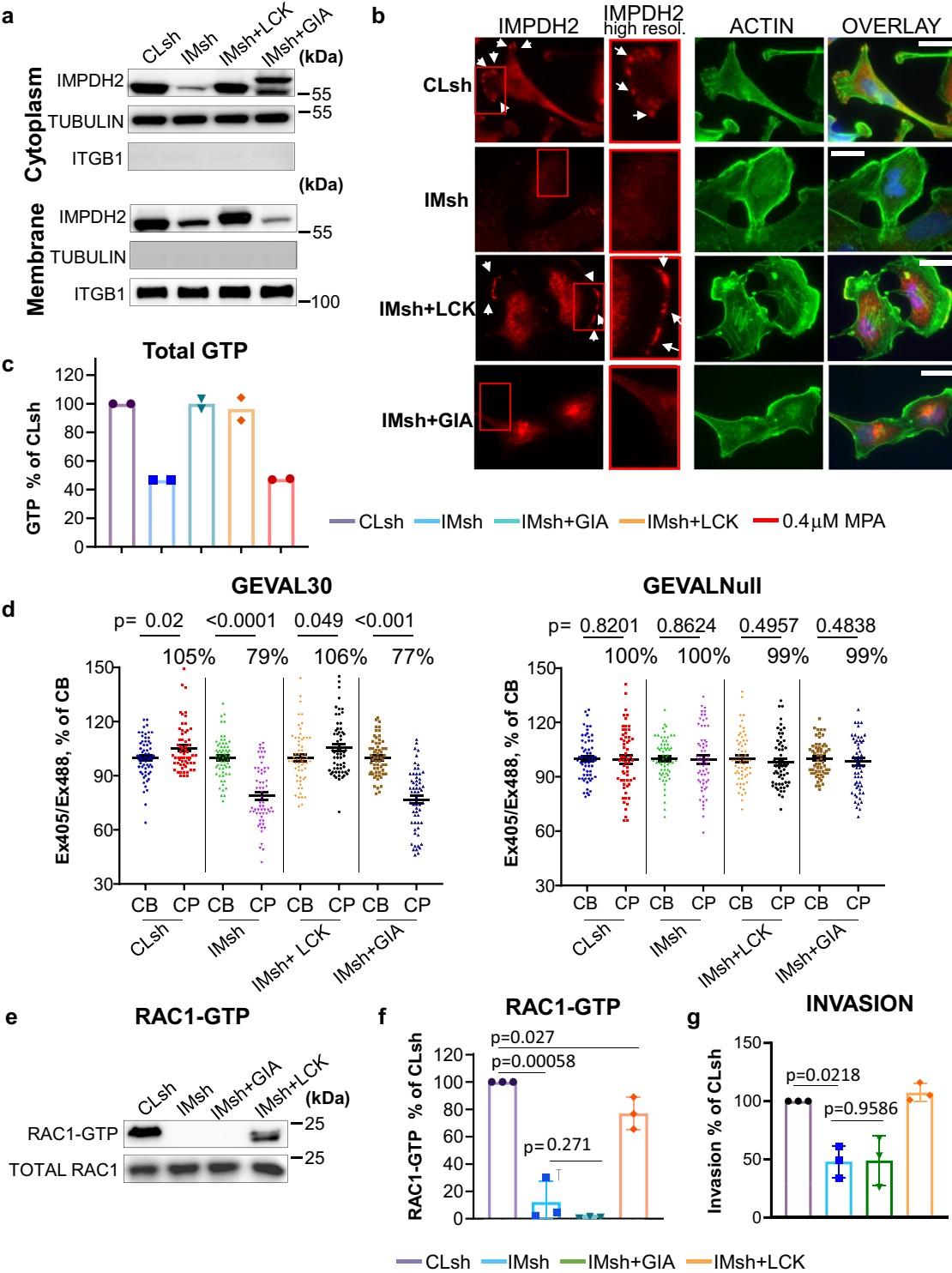

bound IMPDH2 was detected in control cells or cells reconstituted with IMPDH2-LCK but not with IMPDH2-GIA (Fig. 2a, b and Supplementary Fig. 3a).

Depletion of IMPDH2 led to ~50% depletion of total cellular GTP measured via mass spectrometry (Fig. 2c) and ~20% decrease in levels of intracellular free GTP as measured by GEVAL-based assay (Supplementary Fig. 3b), consistent with our previous results[23,24,31]. Rescue of total IMPDH2 levels via overexpression of either IMPDH2-GIA or IMPDH2-LCK constructs restored the total intracellular GTP pool to the levels detected in control cells (Fig. 2c). These data demonstrate that changes in intracellular localization of IMPDH2 do not affect total GTP levels. On the contrary, analysis of GEVAL30 activity in CBs and CPs in the above cell populations demonstrated that rescuing IMPDH2 levels in IMPDH2-depleted cells with IMPDH2-LCK but not with IMPDH2-GIA also restored GTP levels in the CPs of cells (Fig. 2d). Moreover, consistent with these intracellular GTP distribution data, the decrease in RAC1 activity (from whole-cell lysates) produced by IMPDH2 depletion was rescued by expression of membrane-bound IMPDH2-LCK, but not IMPDH2-GIA (Fig. 2e, f).

RAC1 activity has been widely linked to the regulation of cell invasion in cancer. The invasive capability of the MDA-MB-231

**Fig. 2 Intracellular localization of IMPDH2 regulates GTP levels in cell protrusions, RAC1 activity, and cell invasion. a** Cells transduced with the indicated constructs (Cl sh control shRNA, IMsh IMPDH2 shRNA, IM-LCK IMPDH2 with LCK membrane-localization domain, IM-GIA IMPDH2 with giantin Golgi localization domain) were separated into the plasma membrane and cytoplasm fractions as described in Methods, followed by immunoblotting with the indicated antibodies Shown are representative images of at least two independent experiments. **b** Immunofluorescence analysis for IMPDH2 (red) and actin (phalloidin, green), and nuclei (Hoechst, blue) in MDA-MB-231 cells transduced with the indicated constructs. Arrows denote the presence of IMPDH2 at the cell plasma membrane. Scale bar 20 μm. Shown are representative images of two independent experiments. **c** GTP levels were determined via mass spectroscopy as described in Methods. The data represents average ± SEM of two independent experiments performed in duplicates. Statistics performed by two-tailed unpaired Student's $t$-test (****$p < 0.0001$), exact $p$ value <0.00001 for both cases. **d** Quantification of GEVAL activity in cell bodies (CB) and cell protrusions (CP) of MDA-MB-231 cells transduced with the indicated constructs and with GEVAL30 or GEVALNull (30 CBs and 30 CPs per cell type). Horizontal bars represent average. Individual values are from two experiments (15 CB and 15CP per experiment). Statistics were performed by a two-tailed unpaired Student's $t$-test. **e** Cells transduced with the indicated constructs were probed in RAC1 activity assay as described in Methods. Shown are representative images of three independent experiments. **f** Quantification of (**e**). The data represents the average ± SEM of three independent experiments. Statistics were performed by a two-tailed unpaired Student's $t$-test. **g** Invasion assay of MDA-MB-231 cells transduced with the indicated constructs. The data represents the average ± SEM of three independent experiments performed in duplicates. Statistics were performed by two-tailed unpaired Student's $t$-test.

---

cells expressing the above constructs was therefore assessed in an invasion assay with Matrigel-coated Boyden chambers with 8.0 μm pores. IMPDH2 depletion resulted in a 42% reduction in cell invasion (Fig. 2g) consistent with our previous observations[23–25]. Importantly, the invasion could not be rescued by expression of IMPDH2-GIA, (Fig. 3g) despite restoration of total GTP to near wild-type levels (Fig. 2c and Supplementary Fig. 3b). On the contrary, both GTP levels and invasion potential were restored in IMPDH2-depleted cells by expression of IMPDH2-LCK (Fig. 2g). These data suggest that fluctuations in local GTP levels in CPs regulates the invasive capability and RAC1 activity in the studied cells.

**RAC1 activity correlates with intracellular GTP levels.** RAC1 activity in live cells can be monitored using FRET-based sensors, such as our previously described Rac1-FLARE[37–39]. While the original RAC1-FLARE uses wavelengths that overlap with the GEVAL sensors, we have modified its design to red-shift the excitation and emission wavelengths. In the new variant, the mCherry2[40] fluorescent protein was used together with a JF669 far-red dye[41] covalently incorporated using a HALO-tag[42] (Fig. 3a). RAC1-FLARE detects a gradient of RAC1 activation at the leading edge of motile cells[39]. Thus, MDA-MB-231 cells were cotransduced with the RAC1 biosensor and either GEVAL30 or GEVALNull and plated sparsely to induce cell motility. Migrating cells were imaged at short intervals for both RAC1 biosensor and GEVALs activities. In the cells co-expressing RAC1 biosensor and GEVAL30, but not RAC1 biosensor and the GTP-insensitive GEVALNull, visually similar patterns of the activity of both biosensors were observed (Fig. 3b). This observation prompted us to conduct a quantitative correlation analysis of the spatio-temporal distribution of RAC1 activity (as reported by the RAC1 biosensor) and changes in GTP concentration (as reported by GEVAL30). To this end, cells co-expressing RAC1 and GEVAL biosensors were used to simultaneously monitor Rac1 activity and GTP levels at 1-min intervals over 30 min. Pixel-wise correlations between free GTP concentration and Rac1 activity were calculated for each image. A moderate positive correlation was detected between the activities of RAC1 sensor and GEVAL30 (average Pearson correlation coefficient $R = 0.49 ± 0.028$) (Fig. 3c and Supplemental Video 1). In contrast, no correlation was detected between the activities of the RAC1 sensor and GEVALNull ($R = 0.012 ± 0.187$) (Fig. 3c). These data provide independent evidence that RAC1 activity is spatiotemporally associated with the local abundance of GTP at GTP concentration close to GEVAL30 Keff (32.3 μM).

Based on the above results, we were interested in identifying whether changes in intracellular GTP levels would affect activity

of RAC1 proteins with different affinity to GTP. To this end, we utilized previously described RAC1 mutants including RAC1$^{Q61L}$ which has a diminished ability to hydrolyze GTP and remains constitutively bound to it[43]; RAC1$^{P29S}$ which possess increased affinity to GTP due to high GDP/GTP exchange rate with unaffected GTP hydrolysis[44]; RAC1$^{S17N}$ which has approximately tenfold lower affinity to GTP than to GDP compared to wild-type RAC1[45]. HEK293FT cells ectopically expressing the above RAC1 mutants along with wild-type RAC1 were treated with IMPDH inhibitor mycophenolic acid (MPA) that has been previously shown by us to decrease activity of RAC1$^{WT}$[23]. As expected, untreated cells expressing RAC1$^{Q61L}$ or RAC1$^{P29S}$ demonstrated higher amounts of the corresponding GTP-bound RAC1 isoform than RAC1$^{WT}$, whereas the amounts of GTP-bound RAC1$^{S17N}$ were undetectable (in agreement with a published report[45] (Fig. 3d, e). Importantly, treatment with MPA more significantly decreased the amounts of GTP-bound RAC1$^{P29S}$ than RAC1$^{WT}$ ($0.48 ± 0.06$ vs $0.17 ± 0.01$, $p = 0.017$), whereas the amounts of GTP-bound RAC1$^{Q61L}$ remained virtually unchanged (Fig. 3d, e). These data are in agreement with the notion that RAC1$^{P29S}$ cycles GDP/GTP significantly faster than the wild-type RAC1, whereas RAC1$^{Q61L}$ which is incapable of GTP hydrolysis remains in the GTP-bound state. Taken together these data suggest that activity of RAC1 in live cells depends on GTP availability.

**RAC1 directly interacts with IMPDH2.** Cumulatively, the data presented above suggest that RAC1 activity is regulated by the local availability of GTP. Endogenous IMPDH2 was found partially localized at the cell membrane[46] (Fig. 2b), but guanylate metabolism enzymes do not possess a membrane-localization sequence of its own. Thus, we hypothesized that a physical interaction between RAC1 and the GTP-producing enzymes could result in their localization to the membrane, concomitant with regulation of RAC1 via local GTP production. Since IMPDH2 is the rate-limiting enzyme for GTP biosynthesis, we performed immunoprecipitation with control (IgG), IMPDH2-, or RAC1-specific antibodies from MDA-MB-231 and SK-Mel-103 cells followed by an immunoblotting-based analysis of the co-immunoprecipitated materials. IMPDH2 co-immunoprecipitated GMPS, GMPR, NME1, and RAC1 (Fig. 4a and Supplementary Fig. 4). Reciprocally, material co-immunoprecipitated with RAC1-specific antibodies was enriched with IMPDH2, GMPS, GMPR, and NME1 (Fig. 4a and Supplementary Fig. 4 (of note GMPR is not expressed in SK-Mel-103 cells[23] and thus was excluded from the analysis)). Furthermore, shRNA-mediated depletion of IMPDH2 abrogated the ability of RAC1 to co-immunoprecipitate

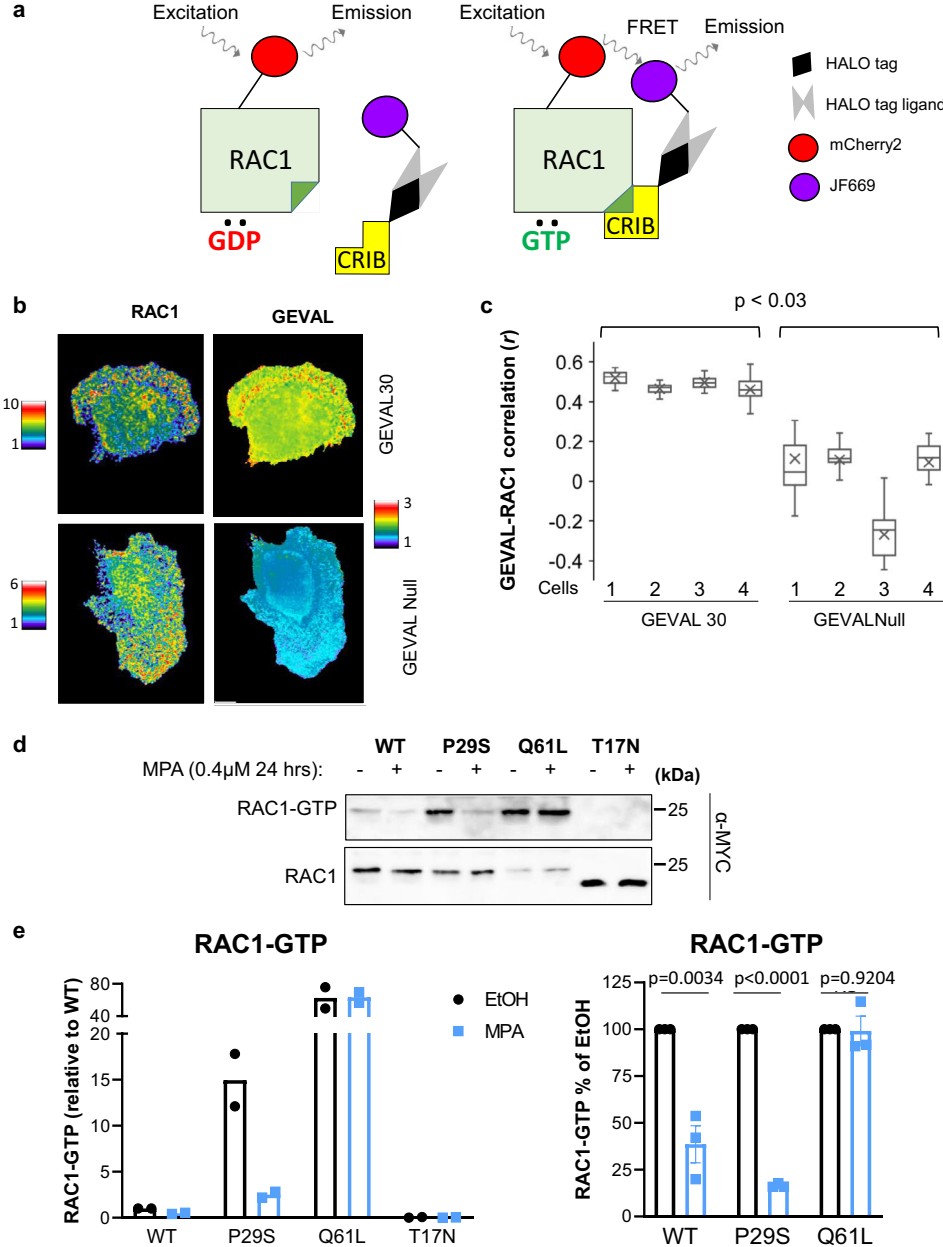

**Fig. 3 Localization of regions with high RAC1 activity and high local GTP concertation. a** Schematic representation of the RAC1 FRET (Förster resonance energy transfer) biosensor. CRIB Cdc42/Rac interactive binding motif. **b** MDA-MB-231 cells co-expressed the RAC1 biosensor and GEVAL30 or GEVALNull. The signal of each individual biosensor was imaged and rendered as described in Methods. **c** MDA-MB-231 cells co-expressed the RAC1 biosensor and GEVAL30 or GEVALNull as in (**b**) and assayed for the correlation between biosensor activities. For each cell, biosensor data were collected in a series of images taken at 1-min intervals over the course of 30 min and analyzed as described in Methods. Pixel-wide Pearson correlation between RAC1 activity (measured as FRET (Förster resonance energy transfer) index for the RAC1 biosensor) and GTP index (measured as the activity of the indicated GEVAL variant) was calculated for each image. The correlation values for the image series corresponding to each individual cell were summarized as "bar-and-whiskers" plots, with "whiskers" indicating the first and the fourth quartiles, the horizontal line (median) splitting the bars into the second and the third quartiles, and "X" indicating quartiles, as well as the mean correlation coefficient (*r*) for each series. The mean correlation coefficients were compared by Mann–Whitney test. **d** Cells expressing the indicated constructs were probed in RAC1 activity assay as described in Methods. Note that the difference in migration of RAC1$^{T17N}$ compared to other proteins is likely due to the fact that RAC1$^{T17N}$ is fused to one Myc-tag, whereas other RAC1 proteins—to two Myc tags. Shown are representative images of at least two independent experiments. **e** Quantification of (**d**); *n* = 2 biologically independent samples (left panel); *n* = 3 biologically independent samples (right panel) The data represents average ± SEM.

GMPS, GMPR, and NME1 whereas depletion of RAC1 did not affect interactions between IMPDH2 and these proteins (Fig. 4a and Supplementary Fig. 4). Furthermore, depletion of RAC1 also decreased the enrichment of IMPDH2, GMPS, GMPR, and NME1 in CPs (Fig. 4b) suggesting that RAC1 at least in part controls localization of these enzymes in the CPs.

To further confirm RAC1–IMPDH2 interactions, we performed a proximity ligation assay (PLA) in sparsely plated migrating MDA-MB-231 cells. This assay allows the in situ detection of protein–protein interactions[47]. The PLA demonstrated strong signals for interactions between RAC1 and IMPDH2 both in cytoplasm and sub-membrane/membrane areas

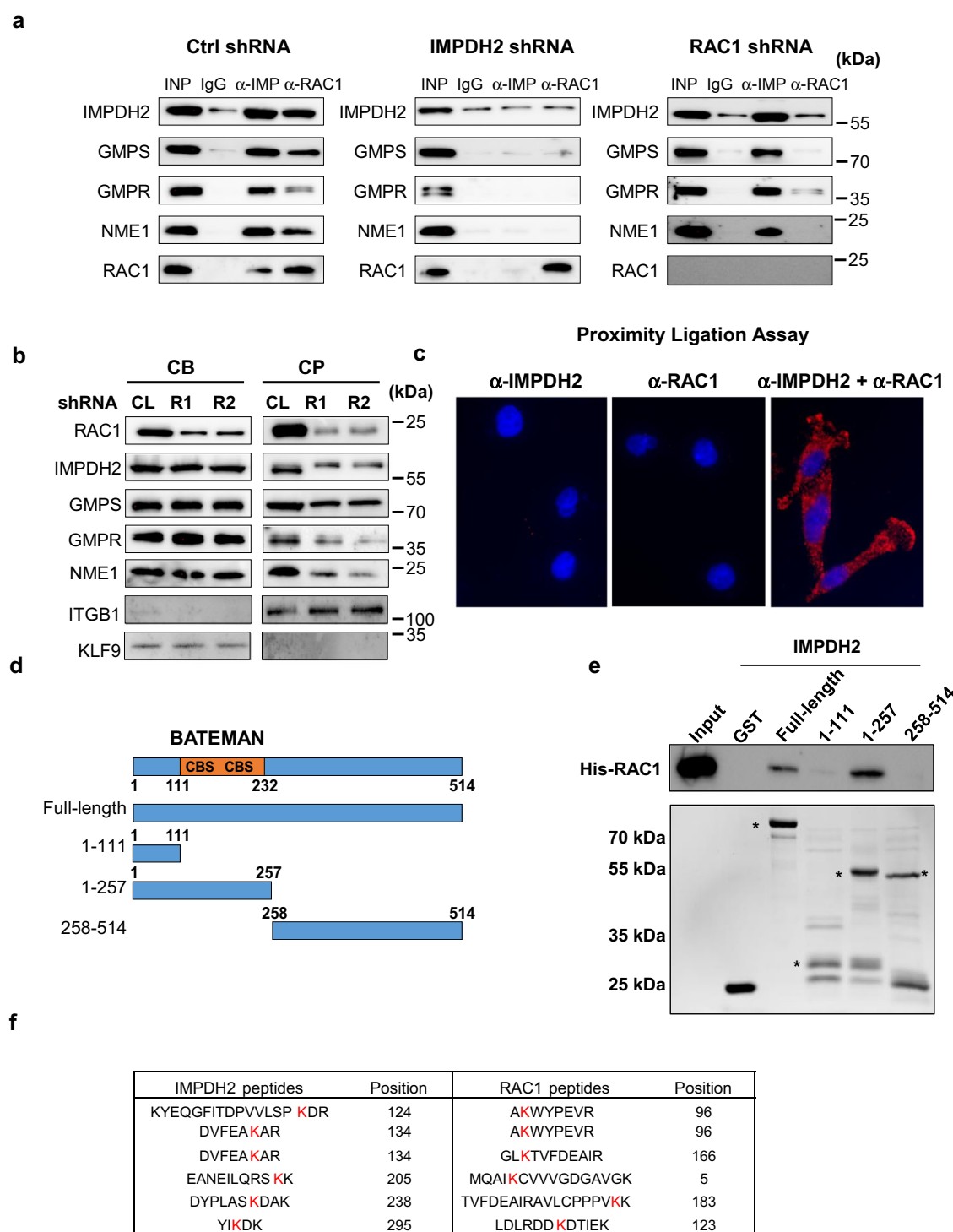

**Fig. 4 RAC1 interacts with IMPDH2 and recruits GMPS. a** Cells were transduced with the indicated constructs and subjected to immunoprecipitation with the antibodies indicated on the top. The immunoprecipitated materials were probed in immunoblotting with the antibodies indicated on the left. Shown are representative images of at least two independent experiments. **b** Cells were fixed with ice-cold methanol and separated into cell bodies (CB) or cell protrusions (CP) fractions, followed by immunoblotting with the indicated antibodies. Shown are representative images of two independent experiments. **c** Proximity ligation assay was performed on sparsely plated cells with the antibodies indicated on the top. Shown are representative images of two independent experiments. **d** Schematic representation of IMPDH2 deletion mutants. Shown are Bateman domain consisting of two cystathionine-β-synthase sequences (CBS). **e** Indicated GST-tagged recombinant IMPDH2 mutants were incubated with recombinant 6xHis-tagged RAC1 (shown on the top), followed by immunoprecipitation with anti-GST tag antibodies and immunoblotting with anti-RAC1 antibodies. Shown are representative images of two independent experiments. **f** Recombinant full-length IMPDH2 and RAC1 proteins were cross-linked followed by mass spectroscopy as described in Methods. Shown are intra-proteins cross-linked peptides identified with highest confidence. Highlighted in red are cross-linked lysines. Shown are representative gel images of at least two independent experiments.

(Fig. 4c) which is in agreement with the previously reported intracellular distribution of RAC1 in mammalian cells[39] and data presented in Fig. 3b and Supplemental Video 1. No signal was detected in PLA control assays with only one primary antibody (RAC1 or IMPDH2) (Fig. 4c). These results suggest that the interaction between IMPDH2 and RAC1 does not dependent on RAC1 translocation to the membrane.

To identify the IMPDH2 domain(s) that mediate direct interactions with RAC1, we generated bacterially expressed recombinant GST-tagged full-length IMPDH2 protein and IMPDH2 deletion mutant proteins (Fig. 4d). In parallel, we generated bacterially expressed recombinant 6xHis-RAC1 protein. RAC1 and IMPDH2-derived proteins were subjected to an in vitro protein binding affinity assay followed by a pull-down using glutathione-agarose beads and detection of RAC1 via immunoblotting (Fig. 4e). The analysis demonstrated that the region of IMPDH2 containing the Bateman domain (111–257aa) (Fig. 4d) was required for IMPDH2–RAC1 interactions. The Bateman domain consists of tandem sequences homologous to cystathionine beta synthase which is present in multiple proteins[48]. This domain is not required for IMPDH2 enzymatic activity[49,50].

To further delineate regions of IMPDH2 interacting with RAC1, we performed chemical cross-linking of recombinant IMPDH2 and RAC1 proteins using disuccinimidyl sulfoxide (DSSO) cross-linking reagent targeting lysines, followed by mass spectrometry analysis, and data analysis by XlinkX[51]. This analysis revealed 39 inter-protein cross-linked peptides (CLPs), among which six high-confidence inter-protein CLPs were identified with an XlinkX score between 10 and 100 (Fig. 4f). XlinkX score is a probability score that describes the confidence of each correctly identified candidate peptide[51]. Importantly, five out of six CLPs identified with high confidence corresponded to the CBS domain region of IMPDH2 (Fig. 4f) which is in agreement with the data described in Fig. 4e, although we could not similarly delineate RAC1 regions interacting with IMPDH2 based on this assay.

## Computer modeling predicts RAC1–IMPDH2 interactions.
To further evaluate RAC1 and IMPDH2 regions involved in the protein–protein interactions, we performed computer modeling using both biased and unbiased approaches. First, we analyzed several structures of RAC1 in complex with other proteins that have been previously determined, including phosphatidylinositol-3, 4, 5-trisphosphate dependent Rac exchange factor 1 (PREX1)[52], Yersinia protein kinase A (YpkA)[53], and DH1 domain of Kalirin[54]. These structures reveal that RAC1 often interacts in a shallow groove between the switch 1 and switch 2 motifs, within the N-terminal 10–76 aa region, recognizing an α-helix in the partner protein (Fig. 5a). The IMPDH2-RAC1 structural model was created by superimposing IMPDH2 into the RAC1-Kalirin complex and aligning the helix containing Lys124 and Lys134 of IMPDH2 within the CBS domain onto the helix of Kalirin within the RAC1 binding site (Fig. 5b). This helix is the most exposed and accessible of the three helices in the Bateman domain of IMPDH2. The model has no major steric clashes and places the RAC1 adjacent to the CBS domain of IMPDH2, so RAC1 binding would not likely interfere with the ability of IMPDH2 to tetramerize. An additional model can be created by positioning the IMPDH2 helix containing Lys206 within the binding groove of RAC1, positioning it near Lys5 of RAC1 (Fig. 4f), however, the N-terminal to C-terminal polarity of the Lys206 helix is opposite of available structures and therefore seems less likely.

In parallel, we applied an unbiased evolutionary coupling (EC) methodology that is based on the assumption that mutation in one of the interacting amino acid residues is likely to be compensated

by a complementary mutation in the other[55]. For predicting the most likely binding interface of the IMPDH2-RAC1 heterodimer, we utilized the direct-coupling analysis (DCA) followed by the Gaussian convolution of EC scores. About 987 and 1189 orthologue sequences of the human RAC1 and IMPDH2 proteins, respectively, were obtained from publicly available databases and analyzed using this methodology. The convolved DCA map of IMPDH2–RAC1 interactions was generated (Fig. 5c) demonstrating that IMPDH2 regions containing 118–137 aa and 156–216 aa (both within the Bateman domain) and 1–52 aa region of RAC1 are the most likely interacting regions. These results are in good agreement with the docking-based prediction of IMPDH2–RAC1 interactions (Fig. 5b) and with the experimental evidence supporting these interactions (Fig. 4d–f). This region corresponds to the GTPases highly conserved switch I and II domain and therefore suggests that IMPDH2 may interact with other GTPases. As a test for this possibility, we probed the material precipitated with IMPDH2 or control IgG for the presence of RHOA and CDC42. The amounts of RHOA were enriched in IMPDH2 but not in IgG co-immunoprecipitated material (Supplementary Fig. 5). Depletion of RAC1 did not substantially affect these interactions. On the contrary, we failed to detect IMPDH2–CDC42 interactions in this assay (Supplementary Fig. 5). Interestingly, our previous data demonstrated that downregulation of total GTP in the cell negatively affects the activity of RAC1 and RHOA but not CDC42[23]. Consistently, depletion of IMPDH2 in MDA-MB-231 cells led to the reduced amounts of active (GTP-bound) RHOA (Supplementary Fig. 6a, b). IMPDH2 depletion did not affect RAS activity, and we did not detect any active CDC42 in these cells (Supplementary Fig. 6a), similarly to previously published work[26].

To empirically evaluate the modeling results, we generated IMPDH2 deletion mutants lacking 101–134 aa or 153–225 aa regions. The cDNAs for IMPDH2 mutants along with wild-type IMPDH2 cDNA were transduced into MDA-MB-123 cells that were depleted of endogenous IMPDH2 as in Fig. 2a (Fig. 5d). All IMPDH2 cDNAs were mutagenized to render them insensitive to the IMPDH2 shRNA as in Fig. 3a. IMPDH2 mutants retained the ability to restore GTP levels in IMPDH2-depleted cells and to precipitate GMPS (Fig. 5d, e). However, neither mutant precipitated RAC1 as efficiently as wild-type IMPDH2 (Fig. 5d) which correlated with the decreased activity of RAC1 in cells expressing these mutants (Fig. 5f, g).

To further delineate the RAC1 region(s) required for interactions with IMPDH2, we performed molecular dynamic (MD) simulations which suggested that 14–25 aa could be critical for binding to IMPDH2 (Supplementary Fig. 7a). MD simulations using mutant RAC1, lacking 14–25 aa, resulted in a higher radius of gyration. Thus, we generated a mutant RAC1 protein lacking 14–25 aa. The RAC1 deletion mutant, wild-type RAC1, both bearing FLAG epitope, and empty vector were transduced into MDA-MB-231 cells followed by immunoprecipitation with anti-FLAG antibodies and probing for IMPDH2 in immunoblotting. Deletion of 14–25 aa negatively affected the ability of RAC1 to co-immunoprecipitate IMPDH2 (Supplementary Fig. 7b) in agreement with the modeling results. On the other hand, RAC1 mutants with different affinity to GTP (Fig. 3d, e) did not substantially differ in the ability to interact with IMPDH2 in co-immunoprecipitation assay (Supplementary Fig. 7c).

Taken together our data demonstrate that interaction with IMPDH2 is important for RAC1 activity.

## Discussion
GTP-binding proteins regulate a vast variety of cellular processes and are frequently observed to be hyper-activated in many human cancers[1,6,56,57]. However, the molecular mechanisms

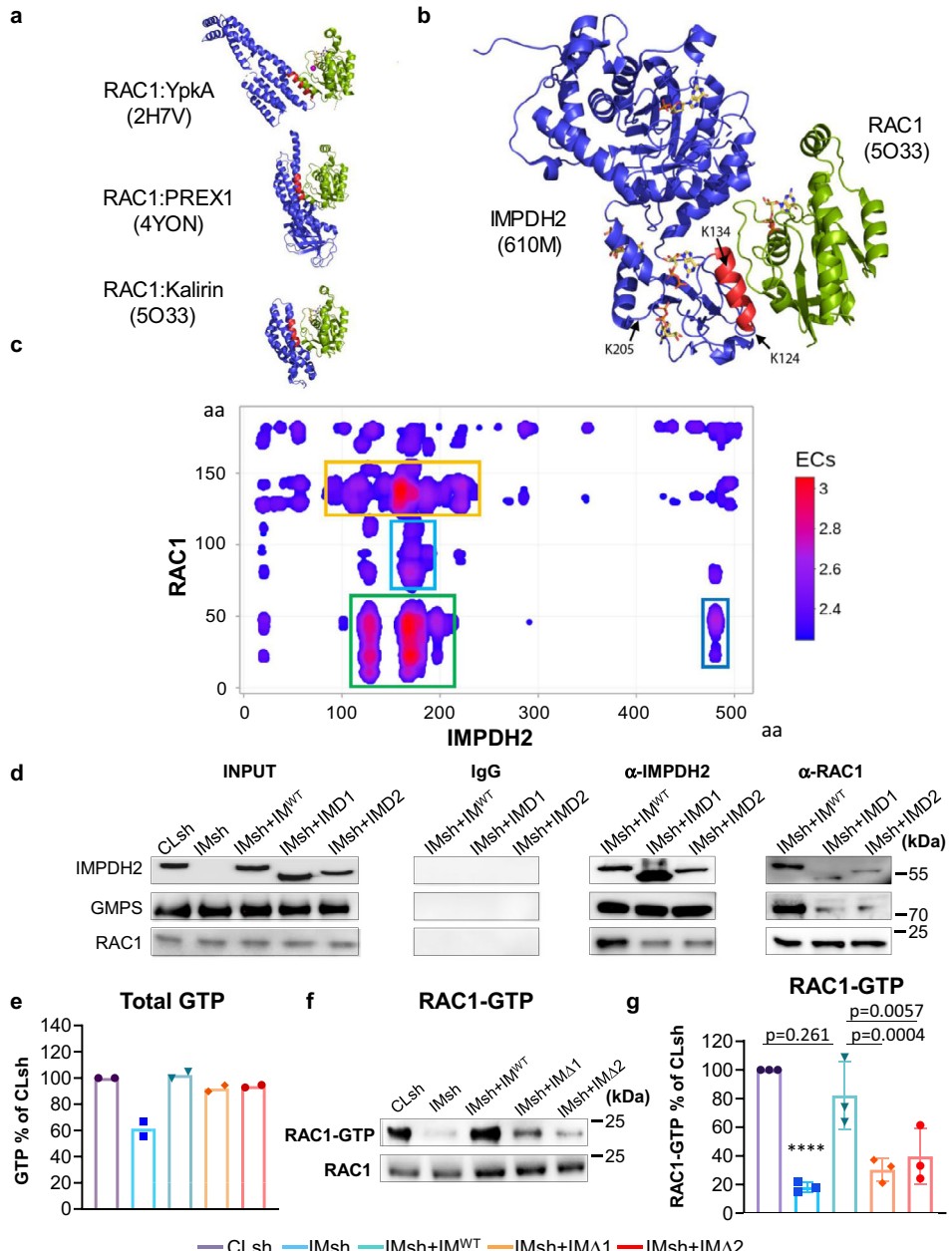

**Fig. 5 Delineation of IMPDH2–RAC1 interactions. a** Structures of RAC1 (shown in green) in complex with other indicated proteins (shown in blue). The number in parenthesis indicates the corresponding entry in RCSB (Research Collaboratory for Structural Bioinformatics) Protein Data Bank. Note that RAC1 binds to α-helices (shown in red) in the target proteins. **b** A model of the IMPDH2–RAC1 interaction was created by superimposing the α-helix of IMPDH2 containing K134 (shown in red) onto the analogous helix in the RAC1-Kalirin complex. **c** Convolved evolutionary coupling of IMPDH2 and RAC1. Strength of the couplings between residues of the two proteins are color-coded with red color denoting the highest probability. Shown in colored frames are regions with a high (green), intermediate (yellow), and low (blue) probability of interactions. **d** Cells were transduced with control shRNA (Cl sh), IMPDH2 shRNA (IMsh), or IMPDH2 shRNA in combination with wild-type IMPDH (IM^WT), IMPDH2 lacking 153-225 aa (IMΔ1), or IMPDH2 lacking 101-134 aa (IMΔ2). Cells were subjected to immunoprecipitation with the antibodies indicated on the top. The immunoprecipitated materials were probed in immunoblotting with the antibodies indicated on the left. Shown are representative images of at least two independent experiments. **e** GTP levels were determined via mass spectroscopy as described in Methods. The data represents the average ± SEM of two independent experiments performed in duplicates. Statistics performed by two-tailed unpaired Student's *t*-test (****$p < 0.0001$), exact *p* value $< 0.00001$. The data represents the average ± SEM of two independent experiments performed in duplicates. **f** Cells transduced with the indicated constructs were probed in RAC1 activity assay as described in Methods. Shown are representative images of three independent experiments. **g** Quantification of (**f**). The data represents the average ± SEM of three independent experiments. Statistics were performed by a two-tailed unpaired Student's *t*-test.

behind their dysregulation are not fully elucidated. Enzymes controlling the de novo biosynthesis of guanylates are also substantially deregulated during tumorigenesis and metastasis[10–16], but their role in the regulation of G-protein activity regulation via

modulation of intracellular GTP levels and thus to cellular transformation is also under-investigated.

The possibility that variations in GTP levels could regulate G-protein activity has been dismissed because in vivo GTP

concentrations were estimated by HPLC or mass spectrometry to be in the 0.5–1.5 mM range[28], which indicate a plentiful reservoir of free GTP. The $K_d$GTP and $K_d$GDP of recombinant RHO-GTPases, measured using non-hydrolyzable guanylate analogs, are very similar and commonly considered to be in the sub-micromolar range[58]. The presence of GEFs generally increases $K_d$GTP of small GTPases (including RHO-GTPases), however, this constant does not exceed the lower micromolar range[29,30], and thus remains much lower than current estimates of in vivo GTP levels. Despite this, we and others reported that moderate changes in GTP intracellular levels affected the activities of several G-proteins including RAC1, RHOA, RHOC, a finding that later confirmed for other small GTPases[23–27]. This discrepancy was hypothesized to be related to the fact that conventional measurements of GTP levels by HPLC or mass spectrometry do not reflect the effective soluble intracellular GTP pool, as GTP may be sequestered in storage forms or bound to abundant GTP-binding proteins or endogenous GTP-binding RNA and DNA aptamers[59], all of which are measured as one pool in the abovementioned methods. These measurements also cannot account for intracellular gradients in GTP distribution.

The data presented here demonstrate that fluctuations of local GTP levels within the range comparable with RAC1 $K_d$GTP play an important role in the regulation of RAC1 activity and cell invasion. Here, using evidence from computer modeling, deletion analysis, reciprocal immunoprecipitation, and PLA, we revealed that RAC1 directly interacts with IMPDH2, recruiting it to cell membrane protrusions that are pivotal for cancer cell invasion. Moreover, our data indicate that by virtue of these interactions, other guanylate metabolism enzymes are recruited to the same locations. These findings further suggest that changes in local GTP levels represent a limiting factor in the regulation of RAC1 activity, which could have important therapeutic implications. Interestingly, physiologically important changes in calcium levels were found using sensors anchored to the membrane[60]. These changes were barely visible, or not at all, with normal calcium sensors. Likewise, the local increases in GTP appear to be similarly restricted and have important physiological roles in the cells.

Both IMPDH2 and RAC1 play important functions in normal cells and simply inhibiting them overall may have unwanted consequences (such as the immunosuppression induced by several IMPDH2 inhibitors[61]). Disabling the interaction between the two proteins to prevent IMPDH2 translocation to the membrane has the potential to offer a more precise suppression of RAC1-mediated invasion. Our results offer a starting point toward these future directions by identifying the interacting domains within IMPDH2 and RAC1.

IMPDH proteins (IMPDH1 and IMPDH2) are homo-tetrameric enzymes that carry out the first and rate-limiting step in the biosynthesis of guanylates. Unlike IMPDH1, IMPDH2 expression is increased in neoplastic cells, and modulation of IMPDH2 levels alters intracellular GTP pools and affects cancer cell proliferation and/or invasion[23–25]. Consistently, in tumor cells used in our study, IMPDH2 mRNA levels were significantly higher than IMPDH1 mRNA levels (Supplementary Fig. 2) although in non-transformed counterparts of these cells the IMPDH2/IMPDH1 level ratio was not as high (Supplementary Fig. 2).

IMPDH enzymes contain a catalytic domain and two evolutionary conserved tandem cystathionine-β-synthase (CBS) domains which form a Bateman domain[20]. CBS domains are found in several proteins, where they play regulatory functions[48]. Likewise, in IMPDH1/2 the Bateman domain is not required for the enzymatic activity but rather for their allosteric regulation[49,50]. Accordingly, our analysis identified that the Bateman domain is required for IMPDH2

interaction with RAC1, and such interactions were predicted to not interfere with IMPDH2 tetramerization. It is noteworthy that other IMPDH2-interacting proteins, such as peptidylprolyl isomerase A (PPIA)[62] and ankyrin repeat domain 9 (ANKRD9)[63], also interact with IMPDH2 via binding to the Bateman domain. At least in the case of PPIA this interaction did not affect IMPDH2 enzymatic activity[62], suggesting that the IMPDH2 Bateman domain may serve as a universal docking module for other proteins. Reciprocally, our computational analysis suggests that 1–52 aa region of RAC1 (that corresponds to switch I and switch II domains conserved in small G-proteins)[64] is involved in binding to the IMPDH2 Bateman domain. This RAC1 region has been previously shown to mediate interaction with other proteins that regulate RAC1 activity, including phospha-tidylinositol-3, 4, 5-trisphosphate dependent Rac exchange factor 1[52], kalirin RhoGEF kinase[54], and dedicator of cytokinesis 9[65]. Our experimental data also suggest that region 14–25 aa of RAC1 which partially overlaps with the switch I domain is important for inter-action with IMPDH2. However, these finding needs to be further evaluated via site-directed mutagenesis to identify key residue(s) directly involved in this interaction since deletions may affect RAC1 overall folding and thus artifactually prevent its binding to IMPDH2. In addition, future experiments utilizing scanning mutagenesis assay and the IMPDH2 mimetic peptides as well as other strategies designed at disrupting the interactions between IMPDH2 Bateman domain and RAC1 will shed light on the exact mechanism of RAC1–IMPDH2 interaction and may offer new options to suppress tumor cells invasion. Due to the fact that IMPDH2 is the most predominately expressed IMPDH isoform in the studied cells, our studies have been focused solely on IMPDH2. However, based on the fact that both proteins share 84% sequence homology which is even higher in the RAC1-interacting Bateman domain and possess comparable enzymatic activities[24], we would anticipate that IMPDH1 interacts with RAC1 and substitutes for IMPDH2 in regulation of RAC1 activity.

We demonstrated that depletion of RAC1 only partially decreased localization of IMPDH2 and GMPS at the CP, suggesting that RAC1-independent mechanisms may be involved. Structural similarities of the switch I–switch II regions among all small G-proteins suggest that IMPDH2 (and other guanylate metabolism enzymes) may be recruited to CP by other RHO-GTPases such as RHOA which interacts with IMPDH2. In addition, other non-yet-identified IMPDH2-interacting proteins may participate in the submembranous localization of IMPDH2. Therefore, a more comprehensive study of these interactions is warranted to better disable IMPDH2 translocation to the cell membrane.

In addition, the recruitment of GTP synthesizing enzymes may serve not only to fuel RHO-GTPase activation but also other proximal GTP-intensive processes, of which the most obvious and demanding may be microtubule dynamics. Thus, RAC1 recruitment of GTP synthesizing machinery may serve to activate and coordinate other GTP-dependent process occurring in CPs during metastasis and cell migration. This may be a general feature in G-protein control of many cellular processes.

Overall, our studies revealed a critical role for local GTP production in the regulation of RAC1 activity and cell invasion (summarized in Fig. 6).

## Methods

**Cell lines**. HEK293FT were purchased from Thermo Fisher Scientific (R70007); MDA-MB-231 cells were purchased from ATCC (CRM-HTB-26). SK-Mel-103 were obtained from the Memorial Sloan Kettering Cancer Center. All cells were cultured in DMEM (Invitrogen, Carlsbad, CA, USA) supplemented with 10% fetal bovine serum and penicillin-streptomycin antibiotics. All cell lines were routinely verified for being mycoplasma-free using the MycoAlert mycoplasma detection Kit purchased from Lonza (Allendale, NJ, USA, Cat # LT07-318). MDA-MB-231 cells

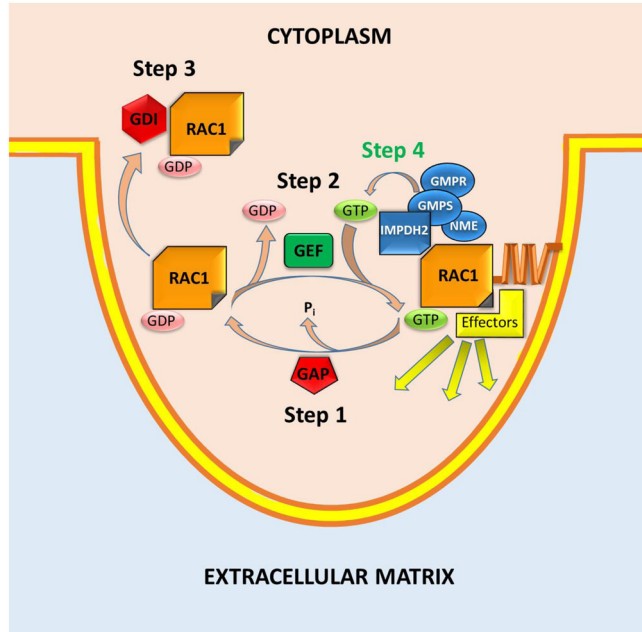

**Fig. 6 Schematic representation of RAC1 regulation by GTP.** GAP GTPase-activating protein, GEF guanine nucleotide exchange factor, GDI GDP dissociation inhibitor.

have been authenticated by STR genotyping at Roswell Park Comprehensive Cancer Center. SK-Mel-103 cells were authenticated at Sloan Kettering Memorial Cancer Center. HEK293T were authenticated at the vendor.

**Immunoprecipitation and immunoblotting**. Immunoprecipitation was performed using the following antibodies: IMPDH2 (Abcam, ab129165), RAC1 (Santa Cruz Biotechnology, sc-514583), Pierce™ Anti-c-Myc Magnetic Beads (Thermo Fisher Scientific, 88842). Immunoblotting was performed with the following antibodies: RAC1 (Proteintech, 24072-1-AP 1:500), GMPS (Abcam, ab228716 1:1000), IMPDH2 (Proteintech, 67663-1 1:500), Tubulin (Proteintech, HRP-66031 1:1000), GMPR (Proteintech, 15683-1-AP 1:1000), NME1 (Proteintech, 11086-2-AP 1:1000), ITGB1 (Proteintech, 26918-1-AP 1:1000), KLF9 ((Santa Cruz Biotechnology, sc-376422 1:250), RHOA (Cell Signaling, #2117, 1:500), CDC42 (Cell Signaling, #8747, 1:200), and RAS (Cell Signaling, #8832, 1:500).

**Plasmids and infection**. The pCMVdeltaR8.2 and pCMV-VSV-G vectors were purchased from Addgene (Cambridge, MA, USA). The pLV-SV4-puro lentiviral vector was obtained from Dr. Peter Chumakov, Cleveland Clinic (Cleveland, OH, USA). shRNAs to RAC1 were purchased from Sigma Aldrich. shRNA to *IMPDH2* was purchased from Sigma Aldrich. shRNA-resistant *IMPDH2* deletion mutants and shRNA-resistant IMPDH2 containing the LCK and Giantin tags at the C-terminus of IMPDH2 were generated using a service from GenScript® (Piscataway, NJ). Lentiviral expression constructs containing FLAG-tagged wild-type RAC1 and RAC1$^{\Delta14-25}$ were generated by VectorBuilder® (Shenandoah, TX). Expression vectors for RAC1$^{WT}$, RAC1$^{P29S}$, RAC1$^{Q61L}$, and RAC1$^{T17N}$ were purchased from Addgene (#128580, #128581, #128582, and #12984, respectively). Lentiviral infections were performed as described in ref. [31].

**Cell fractionation**. Fractionation of plasma membrane and cytosolic fraction was performed using Minute™ plasma membrane protein isolation and cell fractionation kit (Invent Biotechnologies Inc., SM-005) according to the manufacturer's recommendations.

**Immunofluorescence**. Cells were grown on coverslips and fixed in 4% paraformaldehyde (PFA) in PBS. Cells were permeabilized in 0.01% Triton-X100 in PBS and blocked in 3% milk in PBS. Primary antibody hybridization was carried out with a mouse anti-IMPDH2 antibody (cat SAB1406037, Sigma Aldrich, St. Louis, MO) diluted in 1% milk in PBS, and secondary antibody (goat anti-mouse conjugated with AlexaFluor-594, Thermo Fisher Scientific A-11004, 1:500) and AlexaFluor-488-conjugated phalloidin (Thermo Fisher Scientific) staining were carried out in 0.5% milk in PBS. Nuclei were stained with Hoechst 33258 (Thermo

Fisher Scientific). Images were acquired with a Nikon TE2000-E inverted microscope equipped with Roper CoolSnap HQ CCD camera and MetaVue software.

**Generation of the RAC1 activity sensor**. The Rac1 biosensor is a modification of our previously reported dual chain biosensor design [37,38]. To allow simultaneous visualization of Rac1 activities and GTP levels using the GEVAL sensor, we switched the CFP/YFP FRET pair in the Rac1 biosensor for a longer wavelength combination of a red fluorescent protein donor and a far-red dye-based acceptor. The red mCherry2 protein 41 was fused to the N-terminus of full-length Rac1, and a HALO-enzyme that can be labeled with cell-permeable dyes 43 was inserted upstream of residues 60–145 of human PAK1. The two biosensor components were expressed on one open reading frame with two consecutive 2 A viral peptide sequences from Porcine teschovirus-1 (P2A) and Thosea asigna virus (T2A) inserted between them, leading to expression of the two separate biosensor chains with fixed ratio 40, 70. The Rac1 biosensor construct was inserted into a tet-off inducible piggyBac expression system 71 and stable lines were produced in tet-off MDA-MB-231 cells (Johnson Lab, UNC-CH). The GEVAL biosensor was then transduced in these cells using lentivirus. Cells were maintained in DMEM (Cellgro) with 10% FBS (Hyclone) and 0.2 μg/ml doxycycline to repress Rac1 biosensor expression. Rac1 biosensor expression was induced 48 h prior to imaging through trypsinization and washing twice in DMEM/10% FBS and then maintaining in culture media without doxycycline. On the day of imaging, cells were replated using StemPro Accutase (Thermo Fisher Scientific) onto coverslips coated with collagen I (2 μg/ml in 0.02 N Acetic Acid, 37 °C overnight) and allowed to attach in DMEM/10%FBS. After 2 h the cells were labeled with HALO-JF669 (100 ng/ml in culture media, 37 °C 20 min 42 (gift from Dr. Luke D. Lavis, Janelia Research Campus). The cells were then washed twice with imaging media (Hams/ F12 (Caisson Labs) supplemented with 5%FBS, 10 mM HEPES, 100 μm Trolox, and 0.5 mM Ascorbate) for 10 min at 37 °C. The coverslips were then mounted in sealed chambers using degassed imaging media and imaged using a 40×1.3NA Silicon oil objective on an Olympus IX-81 inverted microscope using Metamorph software and 100 W Hg arc lamp illumination. Excitation filters used were FF01-409/32 and FF01-494/20 for the GEVAL sensor, and FF01-561/14 and FF01-650/13 (Semrock) for the Rac1 biosensor combined with a FF410/504/582/669 dichroic (Semrock). For emission, a TuCam dual camera unit (Andor) was fitted with an FF560-FDi01 imaging flat dichroic to separate GEVAL and Rac1 biosensor emissions. The Rac1 biosensor emission was detected using a Gemini image splitter (Hamamatsu) fitted with FF01-615/20 (mCherry2 emission) and FF01-692/40 (JF669/FRET) filters, and an FF640-FDi01 imaging flat dichroic. Images were captured using Flash4 sCMOS cameras (Hamamatsu).

**Individual imaging of GEVAL and RAC1 biosensors**. GEVAL biosensors were designed to detect changes in the amounts of free (unbound) GTP[31]. GEVAL30 was responsive to 4 μM GTP and is started saturating at around 100 μM GTP[31]. The Keff of GEVAL30 (GTP concentrations required to obtain 50% of the maximal ratiometric signal) has been determined as 32.3 μM GTP[31] and GEVAL30 was found to be most responsive to changes in GTP concentrations around its Keff[31]. The GEVALNull sensor served as a critical control to rule out factors unrelated to GTP fluctuation that might affect changes in GEVAL30 activity.

In order to stimulate RAC1 activity and cell protrusion formation leading to cell motility, we plated cells sparsely under normal serum conditions similar to the wound healing assay that induces cell motility. Only motile cells forming CP were imaged for RAC1 and GEVAL analyses. Images from the Rac1 and the GEVAL channels were aligned using fluorescent beads as fiduciaries, through application of the Matlab function "cp2tform" (Matlab, The Mathworks Inc.) to produce a transformation matrix. This was applied to the Rac1 donor image and GEVAL images using the Matlab function "imtransform". For each camera, images obtained without excitation were subtracted from biosensor images to correct for dark current. To correct for shading due to uneven illumination, images of a uniform dye solution, taken under the conditions used for each wavelength, were used to normalize images to an average intensity of 1, producing a reference image for each wavelength. Real images corrected for dark current were divided by the shading correction reference image. Background fluorescence was removed by subtracting, at each frame, the intensity of a region containing no cells or debris. Images were segmented into binary masks separating cell and non-cell regions using the segmentation package "MovThresh"[66], which is based on the Otsu algorithm. The Rac1 donor channel was used for segmentation, as it had the highest signal to noise, particularly at the cell edge. The masks were then applied to all channels, setting non-cell regions to zero intensity.

The GEVAL emission was detected with a FF01-527/20 filter. For GEVAL sensors, activation maps were obtained by dividing the 409ex image by the 494ex image. For the GTPase biosensors, the images were corrected for bleed-through, and ratios were obtained using the following equation (using data from control cells expressing donor or acceptor alone to obtain the bleed-through coefficients α and β): R = (FRET – α(Donor) – β(Acceptor))/donor where R is the Ratio, FRET is the total FRET intensity as measured, α is the bleed-through of the donor into the FRET signal, β is the bleed-through of acceptor into the FRET signal, and Donor and Acceptor are the donor and acceptor intensities as measured through direct excitation. To correct these ratio images for photobleaching, the whole-cell average

was fitted to a double exponential curve and this curve was used to normalize. Pseudocolor scales were produced without considering the lowest and highest 5% of ratio values, to eliminate spurious pixels, and normalized so the lowest value was 1. For the GEVAL sensors, the lowest 5% of ratio values were calculated using the GEVALNull sensor and the high ratio values using the GEVAL30 sensor.

**Correlation analysis of GTP concentration and Rac1 activity**. Each acquired image contained five channels for the five excitation/emission combinations. The fluorescence of the FRET-based RAC1 biosensor was acquired in three channels: "donor" (Ex561/14, Em20/14), "acceptor" (Ex650/13, Em692/40), and "FRET" (Ex561/14, Em692/40). The fluorescence of GEVAL biosensors (GEVAL30 and GEVALNull) was acquired in two channels: Ex409/32 and Ex494/20, both with Em520/15. For each cell, images acquired with 1-min intervals over the course of a 30-min observation were analyzed. For each time-course, the five channels were background-corrected. As ratiometric values are expected to be numerically unstable in areas with little or no fluorophore, pixels falling into the background range in any of the channels were excluded from further analysis. FRET index (an indicator of RAC1 activity) calculations were performed for each scored pixel using the "FRET$_1$" formula in the classification of Berney and Danuser[67]. The GTP index (an indicator of free GTP concentration) for each scored pixel was calculated as the ratio of GEVAL fluorescence in Ex409/32 and Ex494/20 channels. Pixel-wise Pearson correlation coefficients between the FRET index and the GTP index were calculated for each image. The correlation values for the image series corresponding to each individual cell were summarized as "bar-and-whiskers" plots in Microsoft Excel 2016, with options "show mean markers" and "exclusive median" selected. Images in the Supplementary Video were rendered with preset "red hot" (for GEVAL index) and "magenta" (for FRET index) color maps in Fiji software, using [0.45, 0.85] and [0.25, 0.70] scales, respectively.

**Ribonucleotide measurement**. The analysis was performed on a Shimadzu Nexera UHPLC system coupled with a Shimadzu LC-MS-8050 triple-quadrupole mass spectrometer (Kyoto, Japan). All standards were purchased from Sigma (St. Louis, MO, USA) with the exception of UTP purchased from Thermo Fisher Scientific (Alfa Aesar, Thermo Fisher Scientific, Haverhill, MA, USA). An ion-pairing LC-MS/MS method was used to measure ATP, UTP, CTP, and GTP. A mobile phase gradient of ultrapure water (Optima, Thermo Fisher Scientific, Waltham, MA, USA) with 10 mM ammonium acetate (J.T. Baker, Thermo Fisher Scientific, Waltham, MA, USA) and 50 mM tributylamine (Acros Organics, Thermo Fisher Scientific, Fair Lawn NJ, USA) (mobile phase A) and methanol (Optima, Thermo Fisher Scientific, Waltham, MA, USA) with 50 mM tributyla-mine (mobile phase B) was used to separate the analytes. Separation was performed at 0.3 ml/min on a Zorbax Eclipse Plus C18 column (1.8 μm, 2.1 × 100 mm; Agilent, Santa Clara, CA, USA) using the following gradient: 2 min at 0% B, a ramp to 25% B at 8 min, another ramp to 98% B at 12 min, a 3 min hold until 15 min, and then a drop back to 0% B at 15.1 min and allowed to equilibrate there until 25 min. All analytes were monitored in negative mode. The following MRM transitions were used: CTP, 481.90 > 159.00, 481.90 > 384.10; GTP, 521.90 > 159.00, 521.90 > 423.95; UTP, 482.90 > 159.00, 482.90 > 384.90; ATP 505.90 > 159.05, 505.90 > 407.90; and the internal standard for quality control, MES (2-(N-morpholino)ethanesulfonic acid), 194.10 > 80.15, 194.10 > 107.10 m/z.

**Small GTPase activity assays**. RAC1 activity was performed using Rac1 Activation Assay Kit (Cell Biolabs, Inc., STA-401-1), RHOA, CDC42, and RAS activity assays were performed using the corresponding Active GTPase Detection Kit (Cell Signaling, #8820; #8819; and #8821, respectively) according to the manufacturer's recommendations.

**Invasion assay**. After trypsinization and a PBS wash, cells were resuspended at $5 × 10^5$ cells/ml in serum-free DMEM and $2.5 × 10^5$ cells (0.5 ml) were placed in the top compartment of an 8.0 μm BioCoat Matrigel® invasion chambers (BD Bioscience, San Diego, CA, USA) in technical duplicates. DMEM containing 10% FBS was used as a chemoattractant in the lower compartment. Cells were incubated at 37 °C for 18 h, at which time the bottom membranes were fixed and stained with the Hema3 kit (Fisher Scientific, Waltham, MA) according to the manufacturer's protocol. Cells were counted from five different view-fields per transwell.

**CP induction**. Three-micrometer pores transwells were purchased from Thermo Fisher Scientific. A day prior to the experiment, the filters were coated with a 10 μg/ml solution of rat tail collagen type I (Thermo Fisher Scientific) in PBS. For GEVALs experiments, the collagen was also stained with the far-red dye CellTrace Far Red Cell Proliferation Kit (Thermo Fisher Scientific) for visualization during microscopy. Logarithmically growing cells were harvested by trypsinization, washed in PBS, and resuspended in serum-free media (IFM or DMEM, depending on the experimental set-up). About 10% FBS-containing media was placed in the lower chamber as chemoattractant and CP were allowed to form for 2 h at 37 °C. For GEVAL imaging, cells were imaged as described above both above (CB) and below (CP) the collagen-coated filter. In some experiments, a series of Z-stacks was acquired instead of single images. For immunoblot purposes, CB ad CP we fixed in cold methanol on ice for 10 min, rinsed in PBS twice and the selected fraction was collected in radio immune precipitation assay (RIPA) buffer containing proteases inhibitors, after removing the opposite fraction with a cotton swab soaked in PBS and 3x PBS rinses.

**Quantitative real-time PCR**. Total cellular RNA was isolated using the RNeasy Mini Kit (Qiagen, Valencia, CA). cDNA was prepared using cDNA reverse transcription kit (Invitrogen). Quantitative reverse transcription PCR was performed using the following TaqMan™ Gene Expression Assays were used: IMPDH1 = Hs00992210_m1, IMPDH2 = Hs00168418_m1, and ACTB = Hs99999903_m1. PCR data were analyzed using QuantStudio™ Real-Time PCR Software (Thermo fisher).

**Proximity ligation assay (PLA)**. The in situ detection of endogenous protein interaction between IMPDH2 and RAC1 was performed with a Duolink® PLA kit (MilliporeSigma, # DUO92101). MDA-MB-231 cells were seeded in Millicell™ EZ Slides (MilliporeSigma, # PEZGS0816) with overnight culture and then the cell monolayers were scratched with a p200 pipette tip. 12 h after scratching, the cells were fixed with ice-cold methanol for 10 min for PLA assay which was processed according to the manufacturer's instruction. Antibodies used in the PLA assay were rabbit polyclonal anti-IMPDH2 antibody (MilliporeSigma, # HPA001400) and mouse monoclonal anti-RAC1 antibody (Proteintech, #66122). The images were captured using an Olympus IX83 fluorescent microscope and image data was analyzed using Fiji software.

**GST-pull-down assay**. GST-tagged IMPDH2 (wild-type and truncated mutants) and 6xHis-tagged RAC1 were constructed into bacterial expression vectors using PCR-based cloning. The recombinant proteins were purified from IPTG-induced bacterial culture by standard protocols for glutathione-agarose (for GST-tagged IMPDH2 proteins) and Ni-NTA beads (for 6xHis-RAC1). The in vitro protein binding affinity was examined by GST-pull-down assay.

**Protein–protein cross-linking and mass spectroscopy**. DSSO, a mass spectrometry-cleavable and membrane-permeable crosslinker was obtained from Thermo Fisher. In vitro, a DSSO cross-link reaction was performed with the purified recombinant proteins of GST-IMPDH2 and 6xHis-RAC1 following the manufacturer's instruction. The efficiency of cross-link reaction was examined by SDS-PAGE, staining with Coomassie blue R250.

A surfactant-aided precipitation/on-pellet digestion method was used for sample preparation. The reaction mixture after quenching, along with uncrosslinked pure protein (as negative control), was first spiked with 2% SDS to reach a final SDS concentration of 0.5%. Protein was reduced by 10 mM dithiothreitol (DTT) at 56 °C for 30 min and alkylated by 25 mM iodoacetamide (IAM) at 37 °C for another 30 min (in darkness). Both steps were conducted with rigorous oscillation in a thermomixer (Eppendorf). Protein was then precipitated by the addition of 7 volumes of chilled acetone with constant vortexing, and the mixture was incubated at −20 °C for 3 h. After centrifugation at 20,000 x g, 4 °C for 30 min, the supernatant was removed and pelleted protein was carefully rinsed with 500 uL methanol and left to air-dry. The protein pellet was then wetted by addition of 45 uL 50 mM pH 8.4 Tris-formic acid (FA), and a total volume of 5 uL trypsin (Sigma Aldrich) dissolved in 50 mM pH 8.4 Tris-FA was added for 6 h digestion at 37 °C with rigorous oscillation in a thermomixer. Digestion was terminated by the addition of 0.5 uL FA, and digested samples were centrifuged at 20,000 x g, 4 °C for 30 min. The supernatant was carefully transferred to LC vials for analysis.

The LC-MS system consists of a Dionex Ultimate 3000 nano LC system, a Dinex Ultimate 3000 micro LC system with a WPS-3000 autosampler, and an Orbitrap Fusion Lumos mass spectrometer. A large-inner diameter (i.d.) trapping column (300 μm i.d. × 5 mm, Agilent) was implemented before the nano LC column (75 μm i.d. × 65 cm, packed with 2.5 μm Waters XSelect CSH C18 material) for high-capacity sample loading, cleanup, and delivery. For each sample, 4 μL derived peptides were injected for LC-MS analysis. Mobile phase A and B were 0.1% FA in 2% acetonitrile (ACN) and 0.1% FA in 88% ACN. The 180 min LC gradient profile was: 4% for 3 min, 4–11 for 5 min, 11–32% B for 117 min, 32–50% B for 10 min, 50–97% B for 5 min, 97% B for 7 min, and then equilibrated to 4% for 27 min. The mass spectrometer was operated under data-dependent acquisition (DDA) mode with a maximal duty cycle of 3 s. MS1 spectra was acquired by Orbitrap under 120k resolution for ions within the m/z range of 400–1500. automatic gain control (AGC) and maximal injection time was set at 120% and 50 ms, and dynamic exclusion was set at 45 s, ±10 ppm. Precursor ions were isolated by quadrupole using a m/z window of 1.2Th and were fragmented by high-energy collision dissociation (HCD), collision-induced dissociation (CID), or electron-transfer/high-energy collision dissociation (EThcD). MS2 spectra was acquired by Orbitrap under 15k resolution with a maximal injection time of 50 ms. In addition, a CID-OT MS2-HCD-Ion Trap (IT) MS3 data acquisition scheme was adopted for DSSO-cross-linked samples (http://tools.thermofisher.com/content/sfs/posters/PN-64854-LC-MS-Crosslinked-Peptide-IMSC2016-PN64854-EN.pdf)[51,68]. Detailed LC-MS settings and relevant information are enclosed in a previous publication by Shen et al.[69].

LC-MS files were analyzed by Proteome Discoverer 2.2 embedded with XlinkX (Thermo Fisher Scientific). Uncross-linked and CLPs were identified by SequestHT and XlinkX using Swissprot Homo sapiens RAC1 and IMPDH2 sequences as the database. The list of cross-link-spectrum matches (CSMs), cross-linked peptides (CPs), and cross-linking sites (CSs) are exported from Proteome Discoverer and manually curated.

**Docking-based modeling of protein–protein interactions**. The IMPDH2 structure was superimposed onto the GEF Kalirin DH1 domain in complex with RAC1. The backbone atoms of the α-helix containing lysine134 in the CBS domain of IMPDH2 (Pdbid: 6I0M) were superimposed onto the α-helical residues 1364–1374 of Kalirin DH1 domain (pdbid:5O33) using the molecular modeling program, Coot[54]. Structural figures were generated with the program, Pymol (The PyMOL Molecular Graphics System, Version 2.0 Schrödinger, LLC, http:pymol.org).

**Computer simulations**
*Co-evolutionary method for predicting the binding interface of IMPDH2 and RAC1 proteins.* Orthologs of the human RAC1 protein were obtained from 257 vertebrates including birds, alligators, turtles, lizards, mammals, amphibians, coelacanths, bony fishes, and cartilaginous fishes from GENBANK. Additionally, we used tblastn to query the NCBI nonredundant protein database for orthologs not available in GENBANK, retaining of total 987 protein sequences. A similar procedure for IMPDH2 protein produces 1189 orthologous sequences of the human protein. Both sets of orthologous sequences were joined by species and aligned using Clustal-Ω, resulting in concatenated multiple sequence alignments (cMSA) with 961 protein sequences. We then used the direct-coupling analysis method (DCA) to estimate the EC between both proteins as described in Fongang et al.[70]. The pseudo-likelihood maximization direct-coupling analysis (plmDCA) variant of the algorithm that we utilized has a lower computational cost and higher precision than alternative approaches. Options for plmDCA were set as follows: Optimization method (conjugate gradient, cg); Sequence of interest (human, use first member as a sequence of interest); and other parameters as default. As previously mentioned, DCA methods on large proteins may produce a significant number of false-positive, thus we used a Gaussian convolution method to filter out background noise.

To assess the statistical significance of DCA predictions of binding interfaces, we randomly assigned RAC1 sequences to species while keeping IMPDH2 sequences unchanged. The underlined idea was that any randomization of orthologous sequences of one protein will disrupt the phylogeny and remove any evolutionary pressure between the two proteins. Thus, we could test the null hypothesis that the predicted contacts, previously obtained are random and any signal is purely due to noise. To test the null hypothesis, we created 100 randomized cMSA and used the plmDCA algorithm to compute the convolved ECs of each. Then we counted, for each randomized DCA signal, the number of predicted contacts higher than a predefined threshold (minimum ECs score of the top 5% of the original DCA signal). In the case of IMPDH2-RAC1 heterodimer, three randomized DCA signal had their minimum score higher than the threshold, suggesting a score of $3/100 = 0.03$. This score can be rigorously considered as the $p$ value of the interaction given the null hypothesis previously defined. In sum, the IMPDH2–RAC1 interaction as suggested by the predicted binding interfaces, is statistically significant with a $p$ value of 0.03.

*Molecular dynamic method.* MD method was applied to the area of potential interaction of RAC1 with IMPDH2. Amino acids in these regions were then ranked according to their EC scores leading to the hypothesis that the 14–25 aa region of RAC1 could be critical to binding with IMPDH2. Two sets of simulations were performed: (1) simulations with no deletion and (2) simulations with deletion of 14–25 aa of RAC1. The simulations show that the system presents high stability with deletion than no deletion at the beginning of the simulations until about three million time-steps. This is due to the fact that the two proteins are close but do not form a stable complex since the hydrophobic residues are still buried inside the proteins. Upon the multiple conformational changes of both proteins, the hydrophobic residues become exposed to the solvent, which favors the interactions between the two proteins, thus, the formation of a stable complex. After three million time-steps, we observed a stability switch where the wildtype is more stable. The instability of the mutant complex increases with the time-steps, suggesting that the deletion of the 14–25 aa region of RAC1 will lead to the formation of a complex with less stability (weak interactions between both monomers) or a non-formation of the complex.

**Statistical analysis**. Each experiment was performed at least two independent times (the exact number for each experiment is indicated in each figure legend) and the results are expressed as the average ± SEM of the independent experiments unless otherwise noted. Statistical analysis was performed using Student $t$-test. A two-tailed $p$ value <0.05 was considered statistically significant for all analyses.

**Reporting Summary**. Further information on research design is available in the Nature Research Reporting Summary linked to this article.

## Data availability

The experimental data generated in this study are provided in the Source Data. The following databases/datasets have been used in this study: Swissprot Homo sapiens sequences RAC1 (P63000), IMPDH2 (P12268) PDB id: 6I0M, Kalirin DH1 domain PDB is: 5033. GENEBANK. The protein mass spectrometry data generated in this study have been deposited at http://www.proteomexchange.org/ [proteomexchange.org] under accession code PXD028540. Additional information can be found in the associated Life Sciences Reporting Summary. Source data are provided with this paper.

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

## Acknowledgements

We are grateful to Dr. Dominic Smiraglia (Roswell Park Comprehensive Cancer Center) and Dr. Atsuo Sasaki (University of Cincinnati College of Medicine) and for critical reading of the manuscript. This work has been supported by NIH grants CA264984, CA224434, and CA190533 (M.A.N.), CA248018 (A.B.-S.), R35GM122596 (K.M.H), and NS045630 (M.L.F.); T32CA247819 (D.W.W.) The Roswell Park Alliance Foundation grants (to A.B.-S and E.S.K.); and in part by NCI Cancer Center Support Grant P30CA16056 (to the Roswell Park Comprehensive Cancer Center). The authors wish to acknowledge the Wake Forest Baptist Comprehensive Cancer Center Proteomics and Metabolomics Shared Resource, supported by the National Cancer Institute's Cancer Center Support Grant award number P30CA012197.

## Author contributions

M.A.N., A.B.-S., K.M.H. designed the study. D.W.W., Z.D., S.M., Z.H., J.C., R.M.W., A.L.M., D.-H.Y., and A.O'B.C. performed experiments and analyzed data; M.A.N., A.B.-S., K.M.H., J.Q., C.F., and R.J.S. supervised the study and analyzed data. Molecular docking computational analysis was performed by T.H. Evolutionary coupling-based analysis was done by J.B.N.K. and B.F. Mass spec analysis of cross-linked proteins was done by S.S. and supervised by J.Q. GEVAL sensor imaging experiments were done by E.H. and supervised by M.L.F. and A.B.-S. Imaging experiments with cells co-expressing RAC1 and GEVAL sensors were done by D.J.M. and supervised by K.M.H. The data from these experiments were analyzed by D.J.M., M.E.K., and E.S.K. The manuscript was written by M.A.N. and A.B.-S. All authors discussed the results and commented on the manuscript.

## Competing interests

The authors declare no competing interests.

**Additional information**

