## [Peer Review File · Nature Communications]

REVIEWER COMMENTS

Reviewer #1 (Remarks to the Author):

This paper reports the important and truly astounding finding that the activity of the small G protein Rac1 is regulated by fluctuations in GTP levels in mammalian cells.

Rac1 is active when GTP bound and inactive when GDP bound. In prokaryotes, starvation can lower cellular GTP levels to an extent that GTP levels are limiting for the activation of small G proteins. However, homeostatic mechanisms in mammalian cells have always been considered to keep cellular GTP levels so high (mM range) that GTP availability would never be limiting.

This paper shows otherwise.

The authors use a GTP biosensor (GEVAL30) which they recently developed (Ref 35) and measurements of Rac1 activity to show that localised fluctuations in intracellular GTP concentrations determine Rac1 activity, particularly in cell protrusions. In addition, Rac1 is shown to physically interact with enzymes from the GTP biosynthesis pathway, and this interaction is shown to be required for Rac1 activity.

Fig 2: Authors used the GEVAL30 reporter to monitor GTP levels in breast cancer and melanoma cell lines in a transwell system with 3 μ M pores, with FBS as chemoattractant to induce protrusion through the pores. Cell protrusions and cell bodies were compared. GTP levels are shown to be higher in the protrusions than cell body. The authors mechanically separated the cells into body and protrusions, and western blot analysis of the fractions showed that Rac1 and two proteins from the GTP biosynthesis pathway, IMPDH2 and GMPS, are enriched in protrusions.

Fig 3 shows that re-localisation of IMPDH2 to the Golgi (depletion of endogenous with shRNA and rescue using mutants with altered subcellular localisation signals) did not affect total cellular GTP levels but did reduce local GTP levels within cell protrusions, as well as Rac1 activity and the ability of the cells for invasive migration, whereas rescue with an IMPDH2 mutant targeted to the plasma membrane did not.

Fig 4. Authors used a modified Rac1 biosensor (with fluorophores compatible for use in combination with GEVAL30) to show a positive correlation between GTP levels and Rac1 activity over time.

Figs 5/6 shows co-immunoprecipitation of Rac1 with IMPDH2 and GMPS. Proximity ligation was used to show that the interaction of Rac1 with IMPDH2 is likely direct. Mutagenesis, crosslinking and

modelling were used to identify the region of required for interaction of IMPDH2 with Rac1. IMPDH2 mutants deficient in Rac1 binding were generated and shown to be functional in GTP production but causing decreased Rac1 activity.

This is a solid paper which scores top marks for novelty and biological importance. I have queries regarding the biosensor and signalling.

Comments:

1) The authors must explain clearly in the Introduction (or provide data showing) the sensitivity and dynamic range of the GTP biosensor GEVAL30. This information is essential for understanding the levels and localised changes in GTP concentration described here. I looked at ref 35, but I'm still not clear on this: What are the minimum and maximum absolute concentrations of GTP that this sensor can detect? What is the minimal and maximal change it can detect (within one experiment)? Is it possible to translate ratiometric fluorescence values directly into GTP concentration? The authors state repeatedly that GEVAL30 can detect changes of up to 30 μ M, but this statement is not very meaningful without knowing which concentration range it operates in.

2) Fig 3 shows that subcellular localisation of IMPDH2 is important for localised GTP production. The authors should investigate signals that regulate the localisation of IMPDH2, or at least they should speculate on such signals. The interaction with Rac1 is shown to play a large part, but is that interaction constitutive, or does it occur after Rac1 translocates from the cytoplasm to the membrane, or when Rac1 is GTP loaded?

3) Fig 4 shows a positive correlation between GTP levels and Rac1 activity over time. It is not clear why levels would change, as no culture/stimulation conditions were specified. If cells were only observed under basal conditions, the authors should test if cell stimulation (eg with FBS) alters absolute levels and the relationship between GTP levels and Rac1 activity.

Minor

4) Fig 3E/F: specify if Rac activity was measured in whole cell lysate or specifically in the protrusion fraction.

Reviewer #2 (Remarks to the Author):

Technical review comments:

The authors applied an ion-pair LC-MS/MS method to determine intracellular triphosphate nucleotide (NTP) concentrations (including ATP, CTP, GTP and UTP). The method was adequately well described in the manuscript. One concern is that the authors used MES but not stable isotope-labeled NTPs as the internal standard, which may not adequately correct for the influence of matrix effect on ionization efficiency across different cell samples and different NTPs. Regardless, the study focused on the comparison of relative GTP intracellular levels between the cell samples with similar matrix background. Thus, the present method appears to be good enough to serve this purpose.

Two suggestions:

- 1) Please spell out "MES".
- 2) To illustrate the performance of the LC-MS/MS method, please add a supplementary Figure showing the extracted ion chromatograms for ATP, CTP, GTP and UTP in both standard solution and representative cell samples.

Reviewer #3 (Remarks to the Author):

This work addresses the issue of whether the local production of GTP is required for the proper activation and function of the GTPase Rac1 in cell protrusions. The manuscript builds up on previous observations made by the same group indicating that the depletion of enzymes involved in GTP biosynthesis impair the migration of tumor cells. Using biosensors, signaling and biochemical approaches, the authors convincingly show that such a local production of GTP seems to be required for proper Rac1 function. This is a rather breakthrough observation, given that it has been assumed up to now that the high levels of GTP present in cells are enough to promote the activation of the GTPases by the passive incorporation into the nucleotide-free GTPase state. Specifically, the authors show that:

1. Changes in the concentration of GTP do occur in cell protrusions.
2. That the depletion of one of the enzymes involved in GTP production (IMPDH2) leads to low Rac1 GTP-loading and function. This can be rescued by the restoration in those cells of the expression of IMPDH2 with membrane but not Golgi targeting sequences. Likewise, no rescued is observed with a catalytically deficient mutant of IMPDH2.

3. There is a physical interaction between Rac1 and IMPDH2, association that has been only partially dissected.

In general, I find the manuscript interesting. Clearly, it reports observations that can change the view of how Rac1 is activated in cells. I also find that most of the experiments are well performed and most of the data supported by the results provided.

I have, however, some issues that can improve the information about this new regulatory mechanism and its potential impact in the GTPase field:

1. Information about the localization of GMPR (which antagonize IMPDH2 action) and NEM (which eventually promote the generation of GTP) proteins in cell protrusions and the cell body must be given. In the same context, it is also important to show whether all the biosynthetic machinery of GTP co-immunoprecipitates with Rac1 in an IMPDH2-dependent manner.

2. Given that XMP production can be done by either IMPDH2 and IMPDH1, what is the reason for the impact of the IMPDH2 depletion on GTP and Rac1 activation levels?. Is IMPDH1 expressed in those cells? What are the relative expression levels of both isoforms? Does the overexpression of IMPDH1 rescue the phenotype of IMPDH2-knockdown cells? Can IMPDH1 interact with Rac1?

3. What is the state of Rac1 that binds the best to IMPDH2? GTP-Rac1? GDP-Rac1? Nucleotide-free Rac1? All of them?

4. Mutations in the putative IMPDH2 binding sites of Rac1 must be done to further confirm the interaction.

5. What happens with other GTPases, namely RhoA and Cdc42, that must participate in coordinated processes with Rac1? Are they affected by the loss of IMPDH2? Do they also associate with IMPDH2? Are dependent on the IMPDH2-Rac1-interaction? What happens with other GTPases, such as the classical Ras proteins?

6. What happens with the activity of Rac1 mutants that are either GTPase-deficient or rapid cycling?

Other issues:

7. The data representation must be adapted to the Nat Commun editorial style.

8. Fig. 1 could be included as an additional panel of Figure 2 or included as supplemental figure.

We are extremely grateful to the Reviewers for their attention to our work and very thoughtful comments and suggestions that helped us to improve our manuscript. Below please find our detailed point-by-point response to the Reviewers' critique.

REVIEWER COMMENTS

Reviewer #1 (Remarks to the Author):

This paper reports the important and truly astounding finding that the activity of the small G protein Rac1 is regulated by fluctuations in GTP levels in mammalian cells.

Rac1 is active when GTP bound and inactive when GDP bound. In prokaryotes, starvation can lower cellular GTP levels to an extent that GTP levels are limiting for the activation of small G proteins. However, homeostatic mechanisms in mammalian cells have always been considered to keep cellular GTP levels so high (mM range) that GTP availability would never be limiting.

This paper shows otherwise.

The authors use a GTP biosensor (GEVAL30) which they recently developed (Ref 35) and measurements of Rac1 activity to show that localised fluctuations in intracellular GTP concentrations determine Rac1 activity, particularly in cell protrusions. In addition, Rac1 is shown to physically interact with enzymes from the GTP biosynthesis pathway, and this interaction is shown to be required for Rac1 activity.

Fig 2: Authors used the GEVAL30 reporter to monitor GTP levels in breast cancer and melanoma cells lines in a transwell system with 3 μ M pores, with FBS as chemoattractant to induce protrusion through the pores. Cell protrusions and cell bodies were compared. GTP levels are shown to be higher in the protrusions than cell body. The authors mechanically separated the cells into body and protrusions, and western blot analysis of the fractions showed that Rac1 and two proteins from the GTP biosynthesis pathway, IMPDH2 and GMPS, are enriched in protrusions.

Fig 3 shows that re-localisation of IMPDH2 to the Golgi (depletion of endogenous with shRNA and rescue using mutants with altered subcellular localisation signals) did not affect total cellular GTP levels but did reduce local GTP levels within cell protrusions, as well as Rac1 activity and the ability of the cells for invasive migration, whereas rescue with an IMPDH2 mutant targeted to the plasma membrane did not.

Fig 4. Authors used a modified Rac1 biosensor (with fluorophores compatible for use in combination with GEVAL30) to show a positive correlation between GTP levels and Rac1 activity over time.

Figs 5/6 shows co-immunoprecipitation of Rac1 with IMPDH2 and GMPS. Proximity ligation was used to show that the interaction of Rac1 with IMPDH2 is likely direct. Mutagenesis, crosslinking and modelling were used to identify the region of required for interaction of IMPDH2 with Rac1. IMPDH2 mutants deficient in Rac1 binding were generated and shown to be functional in GTP production but causing decreased Rac1 activity.

This is a solid paper which scores top marks for novelty and biological importance. I have queries regarding the biosensor and signalling.

Comments:

1) The authors must explain clearly in the Introduction (or provide data showing) the sensitivity and dynamic range of the GTP biosensor GEVAL30. This information is essential for understanding the levels and localised changes in GTP

concentration described here. I looked at ref 35, but I'm still not clear on this: What are the minimum and maximum absolute concentrations of GTP that this sensor can detect? The authors state repeatedly that GEVAL30 can detect changes of up to 30 μM , but this statement is not very meaningful without knowing which concentration range it operates in. Is it possible to translate ratiometric fluorescence values directly into GTP concentration?

The Reviewer is raising several very important points.

GEVAL sensors were designed to detect changes in the amounts of free (unbound) GTP (Bianchi-Smiraglia et al Nature Methods, ref. 35). In that paper we correlated GEVAL activity directly with GTP concentrations *in vitro* via direct titration of GTP which was not bound to other biological molecules. We found that GEVAL30 is responsive to as low as 4 μM GTP and starts getting saturated at around 100 μM GTP (Supplementary Fig. 5A in ref. 35, WT=GEVAL30). These data demonstrate that GEVAL30 sensitivity being highest at the low end of 4-100 μM range, where changes in GTP levels of a few micromolar can be detected, and least sensitive at the high end where changes of several tens of micromolar are required to cause detectable changes in sensor fluorescence. The K_{eff} of GEVAL30 (GTP concentrations required to obtain 50% of the maximal ratiometric signal) was determined as 32.3 μM GTP (ref. 35, Table 1) which suggests that GTP concentrations around 30 μM are optimal for the detection of changes in the GEVAL30's activity.

Since we do not know the proportion of free GTP in live cells, we were hesitant to exactly translate the actual GEVAL30 ratio values (Ex405/Ex488 ratio) to GTP concentrations in the cell. However, we can draw a conclusion based on which our sensors exhibits the greatest range in fluorescence changes when GTP changes occur near the sensor K_{eff} . The sensitivity and range of the GEVAL30 sensor can be understood from the formula used in ref. 35: $F_n = (K_d + [\text{GTP}] / (K_d + R * [\text{GTP}]))$, where F_n is ratio of emission when excited at 400 vs. 485 (F400/F485), normalized to a value of 1.0 at zero GTP; R is the $1/F_n$ value at saturating GTP, and K_d is the dissociation constant for GTP binding to the sensor. This equation generates simple hyperbolic curves as shown below for the GEVAL30 and GEVAL260 sensor (Ref. 35, Table 1, GEVAL260=P12G (using an R-value of 0.4)).

The sensitivity and range of the GEVAL30 and GEVAL260 sensors adapted from ref. 35.

Inspection of the plot on the left shows that GEVAL30 exhibits large changes in fluorescence as GTP varies from ~ 4 to 100 μM , while GEVAL260 exhibits large changes as GTP varies from ~ 80 to 1024 μM . In particular, GEVAL30 Ex405/Ex485 ratios are calculated to change by 50%, from 1.1 to 1.65, as GTP changes from 6 to 57 μM , while the ratio for GEVAL260 will change by only 10%, from 1.01 to 1.12, across the same range. In live cells, we can quantitatively assess changes in the activity of the GEVAL30 sensor (**ratio of Ex400/Ex488 ratios**) (ref. 35). Thus, we demonstrated that a decrease in the activity of GEVAL30 sensor (but not GEVALNull control sensor) caused by IMPDH2 inhibition correlated with a decrease in total GTP levels which were measured via HPLC after the whole cell lysis (Figures 3 c,d in ref.35).

In the current paper, we used the same methodology (ratio of Ex405/Ex488 ratios) to assess changes in the GEVAL30 activity in different parts of the cell. Since GEVAL30 has the optimal ability to detect GTP at around 30 μM (GEVAL30 K_{eff}), we suggested that the observed changes in GEVAL30 activity corresponded to changes in GTP levels close to 30 μM . The GEVALNull sensor served as a critical control which rules out potentially trivial (i.e. not related to GTP fluctuation) explanations for changes in GEVAL30 activity.

We made several changes in the text including replacing “up-to” to clarify the points raised by the Reviewer in Introduction, as requested (page 4), and Materials and Methods (page 20).

What is the minimal and maximal change it can detect (within one experiment)?

In live cells, we were able to detect an 18% change in GEVAL30 activity within 2hrs of addition of MPA (IMPDH inhibitor), with a maximum change of $\sim 40\%$ between 17hrs and 24hrs (Figure 3e in ref. 35,). These measurements corresponded to a higher decrease in total GTP in lysed cells measured by HPLC (65% and 80%, respectively, Figure 3f in ref.35,). This discrepancy could be due to the difference in the sensitivity between GEVAL30- and HPLC-based methodologies or due to the fact that some intracellular GTP exists in the bound state.

2) Fig 3 shows that subcellular localisation of IMPDH2 is important for localised GTP production. The authors should investigate signals that regulate the localisation of IMPDH2, or at least they should speculate on such signals.

The Reviewer is raising an excellent point. We believe that the detailed investigation of the mechanisms regulating IMPDH2 intracellular localization in addition to Rac1 is a subject for a separate study. However, we speculated about the nature of such mechanisms in the Discussion. Briefly, in the current paper, we have mapped the Rac1 switch I and switch II domains as IMPDH2-interacting region. These domains are conserved in small G-proteins, thus suggesting that other GTPases may fulfill similar function in recruiting IMPDH2. Consistently with this possibility, in the revised manuscript we show that IMPDH2 is able to co-immunoprecipitate RhoA (Supplementary Fig. S5). In addition, unpublished results from our labs show that depolymerization of the actin cytoskeleton drastically reduces the amounts of IMPDH2 at the cell membrane, therefore suggesting the presence of potential additional mechanisms yet to be identified.

The interaction with Rac1 is shown to play a large part, but is that interaction constitutive, or does it occur after Rac1 translocates from the cytoplasm to the membrane, or when Rac1 is GTP loaded?

Data from proximity ligation assay (Fig. 4C) demonstrate that IMPDH2 interacts with Rac1 in the cytoplasm thus suggesting that these interactions do not require IMPDH2 translocation to the membrane. To some extent, data on the interaction between bacterially expressed IMPDH2 and Rac1 also suggest the same.

To address the 2nd question, we utilized Rac1 mutants that have been shown previously to differ in the ability to retain GTP. Those included constitutively active Rac1^{Q61L} which slow hydrolyzes GTP¹ and therefore should not be sensitive to GTP fluctuations; Rac1^{P29S}, which rapidly exchanges GDP/GTP² and therefore, should have increased sensitivity to changes in GTP; and Rac1^{S17N} which has ~10 fold lower affinity to GTP than to GDP compared to the wildtype Rac1³. These Rac1 mutants along with wildtype Rac1 were tested for the ability to interact with IMPDH2 in co-immunoprecipitation and for changes in the activity in response to IMPDH2 inhibition.

We found that RAC1 mutants with different affinity to GTP (Fig. 3DE) did not substantially differ in the ability to interact with IMPDH2 in co-immunoprecipitation assay (Supplementary Fig. S7C).

We also found that untreated cells expressing RAC1^{Q61L} and RAC1^{P29S} demonstrated higher amounts of the corresponding GTP-bound RAC1 isoform than RAC1^{WT}, whereas the amounts of GTP-bound RAC1^{S17N} were undetectable (in agreement with published reports³) (Fig. 3DE). Importantly, treatment with IMPDH2 inhibitor (MPA, that has been previously shown by us to decrease activity of RAC1⁷) more significantly decreased the amounts of GTP-bound RAC1^{P29S} than RAC1^{WT}, whereas the amounts of GTP-bound RAC1^{Q61L} remained unchanged (Fig. 3DE). These data further support the notion that activity of RAC1 in live cells depends on GTP availability.

3) Fig 4 shows a positive correlation between GTP levels and Rac1 activity over time. It is not clear why levels would change, as no culture/stimulation conditions were specified. If cells were only observed under basal conditions, the authors should test if cell stimulation (eg with FBS) alters absolute levels and the relationship between GTP levels and Rac1 activity.

We apologize for not providing sufficient experimental details of the assay. In the original paper describing Rac1 sensor¹, we demonstrated a gradient of Rac1 activation at the leading edge of motile cells. In the current paper, in order to stimulate Rac1 activity gradient and cell protrusion formation leading to cell motility, we plated cells sparsely (under normal serum conditions which contains growth factors stimulating Rac1) similar to the wound healing assay that is being used to induce cell motility. Only motile cells forming cell protrusions and therefore having Rac1 being gradually activated during this process were imaged for Rac1 and GEVAL analyses. This gradient in Rac1 activity enabled us to correlate Rac1 activity with changes in GEVAL30 activity. We made changes in the Result section (pages 7,8) and in Materials and Methods (page 20) to clarify this point.

Minor

4) Fig 3E/F: specify if Rac activity was measured in whole cell lysate or specifically in the protrusion fraction.

The activity of Rac1 was measured in the whole cell lysate as the fixation step needed to obtain the protrusion fraction is incompatible with the active GTPase assay. We have now clarified this point in the revised text (page 7).

- 1 Kraynov, V. S. *et al.* Localized Rac activation dynamics visualized in living cells. *Science* **290**, 333-337 (2000).
- 2 Davis, M. J. *et al.* RAC1P29S is a spontaneously activating cancer-associated GTPase. *Proceedings of the National Academy of Sciences of the United States of America* **110**, 912-917, doi:10.1073/pnas.1220895110 (2013).
- 3 Ridley, A. J., Paterson, H. F., Johnston, C. L., Diekmann, D. & Hall, A. The small GTP-binding protein rac regulates growth factor-induced membrane ruffling. *Cell* **70**, 401-410, doi:10.1016/0092-8674(92)90164-8 (1992).
- 4 Hedstrom, L. IMP dehydrogenase: structure, mechanism, and inhibition. *Chemical reviews* **109**, 2903-2928, doi:10.1021/cr900021w (2009).
- 5 Zimmermann, A., Gu, J. J., Spychala, J. & Mitchell, B. S. Inosine monophosphate dehydrogenase expression: transcriptional regulation of the type I and type II genes. *Advances in enzyme regulation* **36**, 75-84, doi:10.1016/0065-2571(95)00012-7 (1996).
- 6 Nagai, M. *et al.* Selective up-regulation of type II inosine 5'-monophosphate dehydrogenase messenger RNA expression in human leukemias. *Cancer research* **51**, 3886-3890 (1991).
- 7 Wawrzyniak JA *et al.* A purine nucleotide biosynthesis enzyme guanosine monophosphate reductase is a suppressor of melanoma invasion. *Cell Rep* **5**, 493-507, doi:10.1016/j.celrep.2013.09.015 (2013).

Reviewer #2 (Remarks to the Author):

Technical review comments:

The authors applied an ion-pair LC-MS/MS method to determine intracellular triphosphate nucleotide (NTP) concentrations (including ATP, CTP, GTP and UTP). The method was adequately well described in the manuscript. One concern is that the authors used MES but not stable isotope-labeled NTPs as the internal standard, which may not adequately correct for the influence of matrix effect on ionization efficiency across different cell samples and different NTPs. Regardless, the study focused on the comparison of relative GTP intracellular levels between the cell samples with similar matrix background. Thus, the present method appears to be good enough to serve this purpose.

Two suggestions:

1) Please spell out "MES".

MES stands for 2-(N-morpholino)ethanesulfonic acid. This clarification was added to the Materials and Methods.

2) To illustrate the performance of the LC-MS/MS method, please add a supplementary Figure showing the extracted ion chromatograms for ATP, CTP, GTP and UTP in both standard solution and representative cell samples.

These data are now provided in Supplemental Material files.

Reviewer #3 (Remarks to the Author):

This work addresses the issue of whether the local production of GTP is required for the proper activation and function of the GTPase Rac1 in cell protrusions. The manuscript builds up on previous observations made by the same group indicating that the depletion of enzymes involved in GTP biosynthesis impair the migration of tumor cells. Using biosensors, signaling and biochemical approaches, the authors convincingly show that such a local production of GTP seems to be required for proper Rac1 function. This is a rather breakthrough observation, given that it has been assumed up to now that the high levels of GTP present in cells are enough to promote the activation of the GTPases by the passive incorporation into the nucleotide-free GTPase state. Specifically, the authors show that:

1. Changes in the concentration of GTP do occur in cell protrusions.

2. That the depletion of one of the enzymes involved in GTP production (IMPDH2) leads to low Rac1 GTP-loading and function. This can be rescued by the restoration in those cells of the expression of IMPDH2 with membrane but not Golgi targeting sequences. Likewise, no rescue is observed with a catalytically deficient mutant of IMPDH2.

3. There is a physical interaction between Rac1 and IMPDH2, association that has been only partially dissected.

In general, I find the manuscript interesting. Clearly, it reports observations that can change the view of how Rac1 is activated in cells. I also find that most of the experiments are well performed and most of the data supported by the results provided.

I have, however, some issues that can improve the information about this new regulatory mechanism and its potential impact in the GTPase field:

1. Information about the localization of GMPR (which antagonize IMPDH2 action) and NEM (which eventually promote the generation of GTP) proteins in cell protrusions and the cell body must be given. In the same context, it is also important to show whether all the biosynthetic machinery of GTP co-immunoprecipitates with Rac1 in an IMPDH2-dependent manner.

The Reviewer is raising a very interesting point. To address it, we performed the suggested experiments which demonstrated that NME1 and GMPR were enriched in cell protrusions and co-immunoprecipitated with IMPDH2 and Rac1 in MDA-231 cells (Fig. 1D). NME1 was also enriched in cell protrusions and co-immunoprecipitated with IMPDH2 and Rac1 in SK-Mel-103 cells (Fig. 1D). As we reported previously, GMPR is not expressed in the vast majority of melanoma cells including SK-Mel-103 cells. Thus it was excluded from our analysis in these cells.

2. Given that XMP production can be done by either IMPDH2 and IMPDH1, what is the reason for the impact of the IMPDH2 depletion on GTP and Rac1 activation levels?. Is IMPDH1 expressed in those cells? What are the relative expression levels of both isoforms? Does the overexpression of IMPDH1 rescue the phenotype of IMPDH2-knockdown cells? Can IMPDH1 interact with Rac1?

We apologize for not fully explaining our focus on IMPDH2 over IMPDH1 which was based on the expression analysis of these genes in the studied cells. Although IMPDH1 and IMPDH2 share 84% amino acid sequence homology⁴, IMPDH1 has been demonstrated to constitutively express in normal cells, while IMPDH2 levels have been shown to increase in transformed cells^{5,6}. Since both proteins possess the similar molecular weight and migrate at the same level in PAAG, we could not discern their relative expression via immunoblotting. However in our cells, IMPDH2 mRNA levels were 7.56 and 27.0 higher than IMPDH1 (Supplementary Fig. S2) although in non-transformed cells of the corresponding origins, the IMPDH2/IMPDH1 level ratio was 2.12 and 0.19 (Supplementary Fig. S2).

These data are in agreement with several experiments (now included in Fig. 2C) demonstrating that depletion of GTP pools caused by IMPDH2-shRNA was very similar to that caused by treating cells for 24hrs with 0.4 μ M of MPA (an inhibitor of both IMPDH2 and IMPDH1). This dose of MPA was determined to inhibit cell invasion and RAC1 activity without affecting cell proliferation⁷).

Based on the data on IMPDH2 levels and activity, we concluded that it represents the predominant IMPDH isoform in studied cells. Due to these reasons, we did not perform the substitution analysis, however since both proteins share 84% sequence homology (which is even higher in the Rac1-interacting Bateman domain)⁴ and possess comparable enzymatic activities⁶, we would anticipate that IMPDH1 interacts with Rac1 and substitutes for IMPDH2 in regulation of Rac1 activity. We added these points to the Introduction, Results and Discussion sections (pages 3, 4, 6, 15, and 16).

3. What is the state of Rac1 that binds the best to IMPDH2? GTP-Rac1? GDP-Rac1? Nucleotide-free Rac1? All of them?

To address this question we utilized Rac1 mutants that have been shown previously to differ in the ability to retain GTP. Those included constitutively active Rac1^{Q61L} which slow hydrolyzes GTP¹ and therefore should not be sensitive to GTP fluctuations; Rac1^{P29S}, which rapidly exchanges GDP/GTP² and therefore, should have increased sensitivity to changes in GTP; and Rac1^{S17N} which has \sim 10 fold lower affinity to GTP than to GDP compared to the wildtype Rac1³. These Rac1 mutants along with wildtype Rac1 were tested for the ability to interact with IMPDH2 in co-immunoprecipitation.

We found that untreated cells expressing RAC1^{Q61L} and RAC1^{P29S} demonstrated higher amounts of the corresponding GTP-bound RAC1 isoform than RAC1^{WT}, whereas the amounts of GTP-bound RAC1^{S17N} were undetectable (in agreement with published reports³) (Fig. 3DE). Importantly, treatment with IMPDH2 inhibitor (MPA, that has been previously shown by us to decrease activity of RAC1⁷) more significantly decreased the amounts of GTP-bound RAC1^{P29S}

than RAC1^{WT}, whereas the amounts of GTP-bound RAC1^{Q61L} remained unchanged (Fig. 3DE). These data further support the notion that activity of RAC1 in live cells depends on GTP availability.

We also found that RAC1 mutants with different affinity to GTP (Fig. 3DE) did not substantially differ in the ability to interact with IMPDH2 in co-immunoprecipitation assay (Supplementary Fig. S7C).

4. Mutations in the putative IMPDH2 binding sites of Rac1 must be done to further confirm the interaction.

Unlike IMPDH2, Rac1 is a small protein and we wanted to avoid extensive deletions (up-to 30%, as suggested by the computer analysis in Fig. 5c). Thus in the revised manuscript we performed additional molecular dynamic simulations (see Materials and Methods). Briefly, amino acids in the regions of potential interactions with IMPDH2 were ranked according to their EC scores leading to the hypothesis that the 14-25aa region of Rac1 could be critical to binding with IMPDH2. To test this hypothesis, we performed two sets of simulations: (1) simulations with no deletion and (2) simulations with deletion (i.e., deletion of 14-25aa of Rac1). The simulations showed that the system presents high stability with deletion than no deletion at the beginning of the simulations until about three million time-steps (Supplementary Fig. S7A). This is due to the fact that the two proteins are close but do not form a stable complex since the hydrophobic residues are still buried inside the proteins. Upon the multiple conformational changes of both proteins, the hydrophobic residues become exposed to the solvent, which favors the interactions between the two proteins, thus, the formation of a stable complex. After three million time-steps, we observed a stability switch where the wildtype is more stable. The instability of the mutant complex increases with the time-steps, suggesting that the deletion of the 14-25aa region of Rac1 will lead to the formation of a complex with less stability (weak interactions between both monomers) or a non-formation of the complex.

Thus, we generated a Rac1 protein lacking 14-25aa and demonstrated that the deletion of this region resulted in the reduction in the ability of Rac1 to co-immunoprecipitate IMPDH2 (Supplementary Fig. S7B).

5. What happens with other GTPases, namely RhoA and Cdc42, that must participate in coordinated processes with Rac1? Are they affected by the loss of IMPDH2? Do they also associate with IMPDH2? Are dependent on the IMPDH2-Rac1-interaction? What happens with other GTPases, such as the classical Ras proteins?

Although the main focus of our paper is on IMPDH2 interactions with Rac1, we agree with the Reviewer that it will be interesting to assess the ability of IMPDH2 to interact with other RHO-GTPases especially in view of our previous data demonstrating that depletion of GTP levels with mycophenolic acid affected the activity of Rac1 and RhoA but not Cdc42 or Ras in melanoma cells⁷. In the revised manuscript we performed the requested experiments and demonstrated that:

i) Activity of RhoA was affected by IMPDH2 loss (Supplementary Fig. S6). Cdc42 was found inactive in MDA-MB-231 cells (Supplementary Fig. S6, which is in agreement with previously published data⁸). Activity of RAS was not affected within statistical significance (Supplementary Fig. S6).

ii) IMPDH2 co-immunoprecipitated RhoA, but not Cdc42 (Supplementary Fig. S5).

iii) Interactions between RhoA and IMPDH2 were not substantially affected by depletion of Rac1 (Supplementary Fig. S5).

6. What happens with the activity of Rac1 mutants that are either GTPase-deficient or rapid cycling?

Please see the first part of the response to point 3.

Other issues:

7. The data representation must be adapted to the Nat Commun editorial style.

These changes have been made.

8. Fig. 1 could be included as an additional panel of Figure 2 or included as supplemental figure.

Figure. 1 was moved to Supplementary Fig. S1A.

- 1 Kraynov, V. S. *et al.* Localized Rac activation dynamics visualized in living cells. *Science* **290**, 333-337 (2000).
- 2 Davis, M. J. *et al.* RAC1P29S is a spontaneously activating cancer-associated GTPase. *Proceedings of the National Academy of Sciences of the United States of America* **110**, 912-917, doi:10.1073/pnas.1220895110 (2013).
- 3 Ridley, A. J., Paterson, H. F., Johnston, C. L., Diekmann, D. & Hall, A. The small GTP-binding protein rac regulates growth factor-induced membrane ruffling. *Cell* **70**, 401-410, doi:10.1016/0092-8674(92)90164-8 (1992).
- 4 Hedstrom, L. IMP dehydrogenase: structure, mechanism, and inhibition. *Chemical reviews* **109**, 2903-2928, doi:10.1021/cr900021w (2009).
- 5 Zimmermann, A., Gu, J. J., Spychala, J. & Mitchell, B. S. Inosine monophosphate dehydrogenase expression: transcriptional regulation of the type I and type II genes. *Advances in enzyme regulation* **36**, 75-84, doi:10.1016/0065-2571(95)00012-7 (1996).
- 6 Nagai, M. *et al.* Selective up-regulation of type II inosine 5'-monophosphate dehydrogenase messenger RNA expression in human leukemias. *Cancer research* **51**, 3886-3890 (1991).
- 7 Wawrzyniak JA *et al.* A purine nucleotide biosynthesis enzyme guanosine monophosphate reductase is a suppressor of melanoma invasion. *Cell Rep* **5**, 493-507, doi:10.1016/j.celrep.2013.09.015 (2013).
- 8 Kollareddy, M. *et al.* Regulation of nucleotide metabolism by mutant p53 contributes to its gain-of-function activities. *Nat Commun* **6**, 7389, doi:10.1038/ncomms8389 (2015).

REVIEWERS' COMMENTS

Reviewer #1 (Remarks to the Author):

The authors have addressed all my comments satisfactorily. The additional experiments have significantly improved the manuscript. While the underlying mechanism is still incompletely understood, the revised manuscript provides sufficient mechanistic insight for publication in *Nat Commun*, in addition to the very clear novelty and biological significance. I look forward to further publications on underlying mechanism in the future.

Reviewer #2 (Remarks to the Author):

The authors have adequately addressed my concerns.

Reviewer #3 (Remarks to the Author):

The revised version has been significantly improved. In addition, most of the issues requested in the original revision have been addressed. I only have some comments to make:

1. KEY POINT

I am not fully convinced yet about the data regarding the amino acids of Rac1 that mediate the physical interaction with IMPDH2. According to different *in silico* modelling systems, the authors posit that the most N-terminal side of the GTPase (which includes the two switch regions) is involved in this interaction. However, to demonstrate this, they use a mutant version of Rac1 lacking an internal peptide sequence (residues 14-25). There are several problems with this:

(a) This amino acid stretch does not contain any of the two switch regions of Rac1 (it only cuts the most N-terminal end of the switch region).

(b) Such a truncation probably generates an unfolded protein that cannot interact with IMPDH2 or any other Rac1 binding partner.

Due to these problems, the critical residues of Rac1 involved in such an interaction remain ill-defined. I would recommend to resort to other experimental approaches that would allow to keep the structure of the whole GTPase intact. The most appropriate one would be to carry out point mutations in the predicted areas of the interaction. In this regard, the authors have several clues to focus on some specific candidate residues:

(a) The fact that RhoA also binds to IMPDH2, an observation that must help identifying the conserved residues involved in the interaction.

(b) The in-silico modelling results, which suggest that the binding site must be located somewhere in the switch regions.

(c) Previous data regarding the interaction of GDP dissociation inhibitors (GDIs), GDP/GTP exchange factors (GEFs) and proximal effectors with residues located in the two switch regions.

Even without those clues, the authors can carry out a limited scanning mutagenesis approach to pinpoint the surface-exposed residues of Rac1 involved in this interaction.

I would like to point out that, against the authors' hypothesis, the binding interface of Rac1 could be located outside the switch regions. This is because most interactions mediated by those areas are highly influenced by the type of nucleotide bound to the GTPase. For example, the GDIs, GEFs and proximal effectors will only bind to the switch regions when Rac1 is in the GDP-bound, nucleotide-free and GTP-bound form, respectively. By contrast, the authors have found that such an interaction is basically independent on the nucleotide state of the GTPase (Fig. S7C).

2. MINOR POINTS

1. Some figures show statistical calculations with only two independent experimental points (Figures 3E and S6B). This is not proper according to standard statistical criteria. These figures must have at least three independent data points.

2. Lanes 230-231. I believe that the conclusion of this sentence is incorrect. I believe that the effect of MPA on the GTP levels of wild-type Rac1 is probably due to the fact that this protein version must be mostly loaded with GDP in cells.

3. Lanes 227-233. I would explain better to the readers the reason for the higher effect of MPA on Rac1-P29S than on Rac1-Q61L.

4. Lanes 260-261. I disagree. It is unlikely that the interaction of Rac1 with IMPDH2 is constitutive since, in the cytosol, Rac1 must be preferentially bound to Rho GDIs. The only alternative to this is that IMPDH2 and Rho GDIs bind to different regions of Rac1 (an issue discussed in the key point above).

5. Figure S6A. I do see less active H-Ras in this figure. It would be nice to see the second replica, given that similar values have been obtained based on what is shown in Figure S6B. I also understand that the legend to this figure is wrong (Rac1 activity assays?).

We are very grateful for the Reviewer's comments that helped improve our paper. Below please find our point-by-point response.

Reviewer #3 (Remarks to the Author):

The revised version has been significantly improved. In addition, most of the issues requested in the original revision have been addressed. I only have some comments to make:

1. KEY POINT

I am not fully convinced yet about the data regarding the amino acids of Rac1 that mediate the physical interaction with IMPDH2. According to different in silico modelling systems, the authors posit that the most N-terminal side of the GTPase (which includes the two switch regions) is involved in this interaction. However, to demonstrate this, they use a mutant version of Rac1 lacking an internal peptide sequence (residues 14-25). There are several problems with this:

(a) This amino acid stretch does not contain any of the two switch regions of Rac1 (it only cuts the most N-terminal end of the switch region).

(b) Such a truncation probably generates an unfolded protein that cannot interact with IMPDH2 or any other Rac1 binding partner.

Due to these problems, the critical residues of Rac1 involved in such an interaction remain ill-defined. I would recommend to resort to other experimental approaches that would allow to keep the structure of the whole GTPase intact. The most appropriate one would be to carry out point mutations in the predicted areas of the interaction. In this regard, the authors have several clues to focus on some specific candidate residues:

(a) The fact that RhoA also binds to IMPDH2, an observation that must help identifying the conserved residues involved in the interaction.

(b) The in-silico modelling results, which suggest that the binding site must be located somewhere in the switch regions.

(c) Previous data regarding the interaction of GDP dissociation inhibitors (GDIs), GDP/GTP exchange factors (GEFs) and proximal effectors with residues located in the two switch regions.

Even without those clues, the authors can carry out a limited scanning mutagenesis approach to pinpoint the surface-exposed residues of Rac1 involved in this interaction.

I would like to point out that, against the authors' hypothesis, the binding interface of Rac1 could be located outside the switch regions. This is because most interactions mediated by those areas are highly influenced by the type of nucleotide bound to the GTPase. For example, the GDIs, GEFs and proximal effectors will only bind to the switch regions when Rac1 is in the GDP-bound, nucleotide-free and GTP-bound form, respectively. By contrast, the authors have found that such an interaction is basically independent on the nucleotide state of the GTPase (Fig. S7C).

We agree with the Reviewer's points and made according changes in the text (page 15, last paragraph).

2. MINOR POINTS

1. Some figures show statistical calculations with only two independent experimental points (Figures 3E and S6B). This is not proper according to standard statistical criteria. These figures must have at least three independent data points.

The requested changes were made.

2. Lanes 230-231. I believe that the conclusion of this sentence is incorrect. I believe that the effect of MPA on the GTP levels of wild-type Rac1 is probably due to the fact that this protein version must be mostly loaded with GDP in cells.

We took into consideration the Reviewer's suggestion and made a less assertive statement in the text: "Taken together these data suggest that activity of RAC1 in live cells depends on GTP availability." (page 9, 1st paragraph).

3. Lanes 227-233. I would explain better to the readers the reason for the higher effect of MPA on Rac1-P29S than on Rac1-Q61L.

The explanation is now provided (page 9, 1st paragraph)